# A positive feedback loop between sensory and octopaminergic neurons underlies nociceptive plasticity in *Drosophila* larvae

Jean-Christophe Boivin[1,2], Yi Q. Zhao[1], Jiayi Zhu[1,2], Jared T. Dakin[1], Jing Ning[1], Tomoko Ohyama[1,2,3]*

**1** Department of Biology, McGill University, Montreal, Quebec, Canada, **2** Integrated Program of Neuroscience, Pine Ave. W., McGill University, Montreal, Quebec, Canada, **3** Alan Edwards Center for Research on Pain, McGill University, University St., Montreal, Quebec, Canada

* tomoko.ohyama@mcgill.ca

## Abstract

Adaptive modulation of nociceptive behaviour based on prior experience is essential for responding effectively to environmental threats. In *Drosophila* larvae, nociceptive escape behaviours are robust and stereotyped, yet emerging evidence suggests this can be modulated by experience and internal state. Here, we demonstrate that repeated activation of nociceptive sensory neurons enhances both the likelihood and intensity of nocifensive rolling, reflecting a form of behavioural sensitization. This heightened responsiveness is accompanied by a sustained increase in activity within nociceptive sensory neurons, suggesting that plasticity arises, at least in part, within the sensory compartment. We identified the neuromodulator octopamine as a critical regulator of the sensitization: signalling through the octopamine receptor OAMB is required to sustain elevated nociceptive gain, and feedback from one of octopaminergic neurons class, the ventral unpaired median (VUM) neurons, amplifies sensory neuron output. Together, these findings reveal an experience-dependent positive feedback loop in the nociceptive system, where neuromodulatory circuits tune behavioural output.

### Author summary

The ability to adapt behavior based on past encounters with danger is a fundamental survival trait. Repeated exposure to harmful or painful stimulus often leads animal respond more strongly over time, process known as sensitization. Although sensitization is critical for avoiding future threats, the neural circuitry that allows experience to fine-tune sensitivity is not fully understood.

In this study, we used fruit fly (*Drosophila*) larvae to investigate how repeated noxious stimuli alter behavior. We found that after multiple exposures, larvae are much more likely to perform a characteristic escape rolling behavior, and they

**Data availability statement:** All relevant data are within the paper and its Supporting information files. Original behavior data and imaging data in this manuscript are deposited in https://doi.org/10.5683/SP3/GNIVCE.

**Funding:** This work was supported by McGill University, the National Sciences and Engineering Research Council (NSERC, RGPIN/04781-2017 to TO, training grant to JCB), the Canadian Institute of Health Research (CIHR, PTJ-376836 to TO), the Fonds de recherche du Québec - Nature et technologies (FRQNT, 2019-N-25523 to TO, training award to JCB), Fonds de recherche du Québec - Santé (FRQS, NeuroNex 2019-295825 to TO) the Canada Foundation for Innovation (CFI, CFI365333 to TO). The funders had no role in study design, data collection and analysis, decision to publish, or preparation of the manuscript.

**Competing interests:** The authors have declared that no competing interests exist.

do so with greater intensity. This increased sensitivity arises from changes within the pain-sensing neurons themselves, which stay "on alert" and increase their activity. We identified the neuromodulator octopamine and its receptor, OAMB, as key regulators of this process. Our results reveal that specific set of feedback neurons amplifies pain signalling through positive feedback loop. Together, these finding provide insight into how neuromodulator feedback circuits enable nervous system to adapt behavioral response to environmental threats.

## Introduction

The ability to modify behaviour based on experience is essential for survival in complex and ever-changing environments. This behavioural flexibility arises from the intricate interplay between an organism's genetically determined neural architecture ("nature") and the influence of external conditions and past experiences ("nurture"). Unraveling how these factors interact to produce adaptive behavioural outcomes remains a central goal in neuroscience, requiring a deep understanding of the molecular, cellular, and circuit-level processes that mediate experience-dependent change.

Decades of research have explored the biological mechanisms that support experience-dependent behavioural change, with synaptic plasticity emerging as a central process—particularly in the formation of associations between stimuli or events—across both vertebrate and invertebrate model systems [1–3]. In parallel, technological advances such as high-resolution electron microscopy and optogenetics have enabled detailed mapping and manipulation of entire neural circuits in small brains [4–6]. These tools have opened new opportunities to investigate how specific patterns of connectivity and activity give rise to behavioural outcomes [4–6]. Notably, such studies have highlighted the importance of neuromodulators—chemical signals released more broadly and over longer timescales than classical synaptic neurotransmitters—in dynamically regulating circuit function and enhancing behavioural adaptability [7–9].

Neuromodulators such as dopamine, serotonin, and various neuropeptides play a pivotal role in shaping the functional architecture of neural circuits. In contrast to fast synaptic transmission, neuromodulatory signals exert temporally and spatially diffuse effects, modulating synaptic strength, circuit excitability, and overall behavioural state [7,10,11]. These modulators are uniquely positioned to act as molecular bridges between an organism's genetic blueprint and its experiential history, biasing neural computations in a context-dependent manner [12,13]. Despite increasing recognition of their significance, the precise mechanisms through which neuromodulators drive experience-dependent behavioural plasticity remain incompletely understood.

Invertebrate model systems offer powerful platforms for dissecting these pathways, particularly due to their relatively simple and well-characterized nervous systems. In *Drosophila melanogaster*, the biogenic amine octopamine serves as a major neuromodulator and is widely considered the functional analogue of noradrenaline in mammals. Octopamine modulates a broad spectrum of physiological and behavioural

processes across developmental stages, including locomotion, reproduction, aggression, sleep, metabolism, and various forms of behavioural plasticity [14–20]. Its effects are mediated through a suite of G-protein-coupled receptors (GPCRs), including OAMB, Octα2R, and three OctβRs, which activate distinct intracellular signaling cascades and influence diverse neural substrates [21]. By targeting specific circuits in a context-sensitive manner, octopaminergic signaling plays a central role in aligning behavioural output with environmental demands and internal state [19,20,22,23].

Building on this foundation, the *Drosophila melanogaster* larval system offers a valuable context for examining how neuromodulatory influences shape behavioural responses to ecologically relevant challenges. As a species that occupies a wide range of geographic regions, *Drosophila* larvae are routinely exposed to diverse and often adverse environmental conditions [24]. During development, they encounter various noxious stimuli—including chemical irritants, extreme temperatures, and mechanical injury from parasitic wasps—that threaten their survival [25–29]. To cope with such threats, larvae rely on a well-characterized nociceptive system, in which class IV dendritic arborization (C4da) sensory neurons detect damaging stimuli and drive stereotyped escape behaviours through downstream motor circuits [28–40]. While these responses are robust, accumulating evidence suggests they are not hardwired reflexes; rather, they exhibit substantial plasticity shaped by prior experience. This modulation, which includes both sensitization and habituation, is influenced by factors such as the developmental stage of the larva [41,42], the type of noxious input [35,43,44], and critically, the neuromodulatory state of the nociceptive network [33,37,43,45–48]. However, the interplay between these variables remains poorly understood. Gaining a comprehensive view of how intrinsic and extrinsic factors converge to shape plasticity in nociceptive behaviour is essential for understanding how adaptive or maladaptive responses to environmental threats are sculpted over time—particularly under conditions of repeated or unpredictable stimulation [49–52].

Here, we investigate how nociceptive experience dictates both the magnitude and directionality of nociceptive adaptation and show that increased nocifensive behaviour correlates with enhanced sensitivity to noxious stimuli, which in turn is sustained by elevated activity within nociceptive sensory neurons. Using RNA interference, we demonstrate that the octopamine receptor OAMB is required for experience-dependent sensitization. Moreover, octopaminergic feedback to nociceptive sensory neurons is both necessary and sufficient for this form of plasticity. Specifically, octopaminergic ventral unpaired median neurons (VUMs) are critical regulators of this feedback loop. Together, these findings reveal that experience-dependent sensitization is driven by an octopaminergic positive feedback mechanism between sensory neurons and their downstream modulatory partners.

## Results

### Previous noxious experiences modulate responses to noxious stimuli

A previous study in *D. melanogaster* larvae reported that continuous noxious chemical stimulation of class IV dendritic arborization (C4da) neurons results in behavioural habituation [43]. To investigate how different stimulation patterns might influence the larval nociceptive system and behaviour, we employed optogenetic activation of C4da neurons during development. Larvae expressing CsChrimson in C4da neurons were exposed to 620-nm LED (5.6 µW/mm$^2$) for five seconds every five minutes from the embryo stage until the late third instar stage (Fig 1A). We then assessed nocifensive behaviour by optogenetically activation of C4da neurons and quantifying individual larval response–especially rolling– using machine-learning-based software [53,54] (Fig 1B). Larvae that were developmentally simulated for 120 hours exhibited a markedly enhanced nocifensive rolling compared naive controls (Fig 1B). While only 42% of naïve larvae responded with rolling, 98% of developmentally stimulated larvae displayed rolling behaviour (Fig 1C). Furthermore, the latency to initiate rolling was significantly reduced in the stimulated group, occurring 3.63 ± 0.52 seconds earlier than in naïve larvae (stimulated: 0.67 ± 0.03 s; naïve: 4.30 ± 0.49) (Fig 1D). Once initiated, rolling persisted for a significantly longer duration in stimulated larvae, who spent an average of 10.11 ± 0.31 s engaged in rolling during the 30-second stimulation period, compared to 2.73 ± 0.30 s in naïve animals (Fig 1E). These findings indicate that repeated nociceptive stimulation of

PLOS Genetics

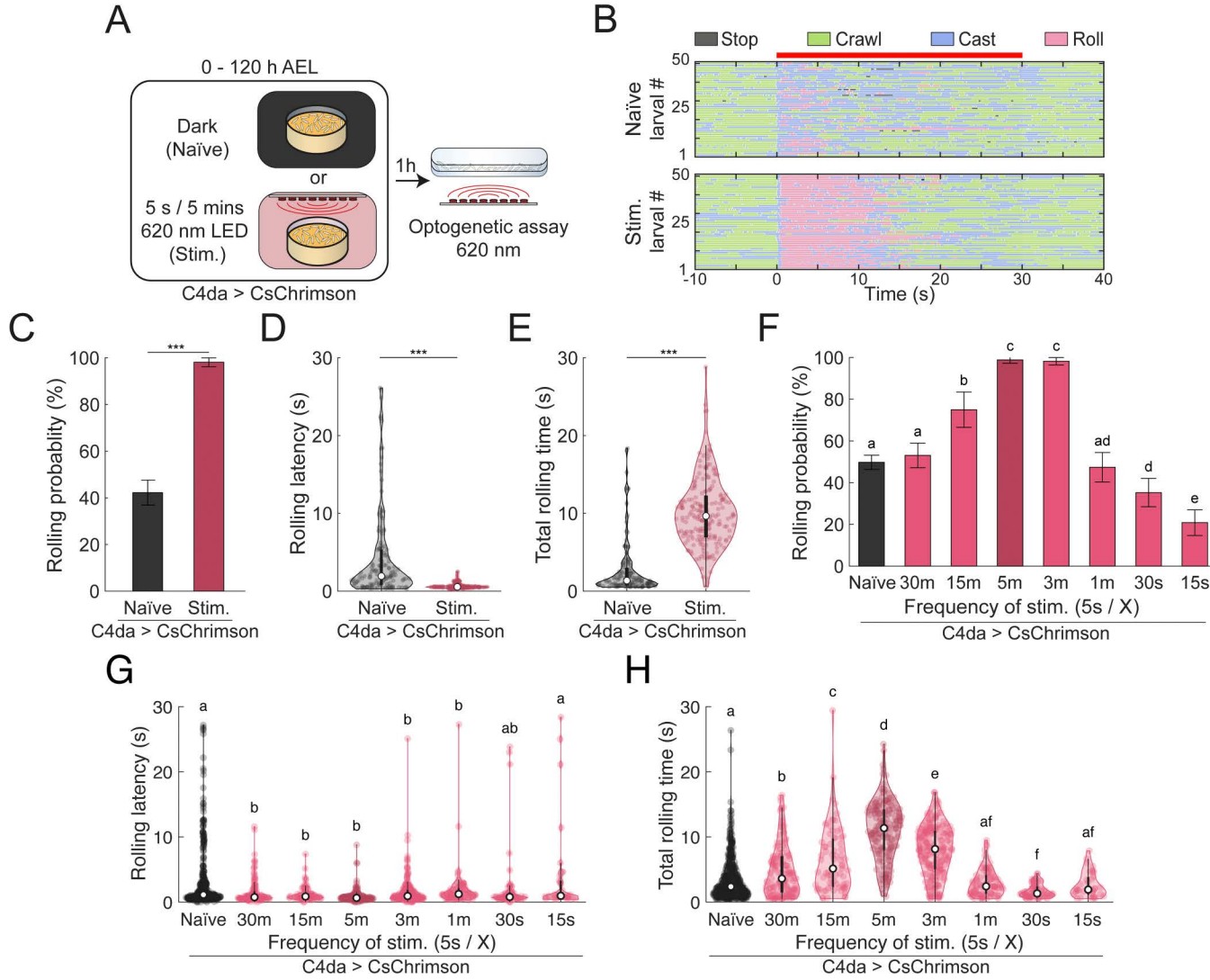

**Fig 1. Noxious experience during development shapes nociceptive behaviours depending on timing and frequency. (A)** Schematic of the experimental paradigm. Larvae were raised on food containing all-transretinal and allowed to develop for 120 hours until late third instar. Larvae were either kept in the dark (Naïve) or subjected to intermittent optogenetic activation of C4da nociceptive neurons during development (Stim.). Behavioural responses were tested via optogenetic stimulation at the late third instar stage. **(B)** Raster plots showing individual behavioural responses (stop, crawl, head cast, roll) over 30 s of optogenetic C4da stimulation. Each row represents a single larva that was tracked continuously for the duration of the stimulation. Top - Naïve larvae. Bottom - Developmentally stimulated larvae. **(C)** Rolling probability over the 30 s stimulation window. Bars indicate mean; error bars denote 95% confidence intervals (n = 327, 207). Chi-square test, ***$p < 0.001$. **(D)** Violin plot showing the rolling latency of larva during optogenetic stimulations (n = 138, 203). Welch's ANOVA Test ***$p < 0.001$. **(E)** Violin plot of total rolling time in response to optogenetic C4da stimulation (n = 138, 203). Welch's ANOVA Test ***$p < 0.001$. **(F-H)** Stimulation frequency dictates directionality of behavioural changes. **(F)** Rolling probability as a function of stimulations during development. Error bars indicate the 95% confidence interval (n = 818, 277, 100, 174, 219, 194, 190, 168). Chi-square test with compact letter display; statistical significance at $p < 0.01$. **(G)** Violin plot of rolling latency during optogenetic stimulation (n = 407, 147, 75, 172, 215, 92, 67, 35). Kruskal-Wallis followed by pairwise Mann-Whitney test, CLD denotes $p < 0.01$. **(H)** Violin plot of total time each larvae spent rolling during optogenetic stimulations (n = 106, 125 138, 113, 165, 229, 254). Kruskall-Wallis followed by pairwise Mann-Whitney test; CLD denotes $p < 0.01$.

C4da neurons during development induces behavioural sensitization, leading to heightened response and prolonged engagement in nocifensive behaviour.

To investigate how the pattern of nociceptive sensory neuron stimulation during development influences nocifensive behaviour, we varied the stimulation parameters, including frequency, light intensity, and developmental timing. We first examined whether acute nociceptive stimulations lead to an increase in nociceptive behaviours as to understand the timescale at which these changes operate. To this end, larvae were repeatedly stimulated for one hour at a frequency of 5s/ 5m. Here, the stimulations lead to a modest increase in rolling probability and a decrease in rolling latency (S1A-S1B Fig). However, there was no significant change in total rolling time (S1C Fig). This suggests that while experience can have an acute effect on subsequent behaviours, the chronicity of the stimulations could strengthen the effect. Then, we looked at the effect of 24-hours stimulation across different developmental stages, comparing against naïve behaviour and stimulations over the entire development. All larvae were tested in their late-third instar (i.e., at 120h AEL). Larvae stimulated during first instar stage were not more likely to roll than naïve controls but showed a reduction in their rolling latency (S1D-S1E Fig). Stimulation during the second instar stage further enhanced sensitization, leading to both decreased latency and increased rolling probability (S1D-S1E Fig). Stimulation during the third instar (i.e., 72h-96h and 96h-120h AEL) resulted in the most pronounced sensitization, with significantly increased rolling probability, reduced latency and prolonged rolling duration compared to naïve animals (S1D-S1F Fig). While no individual 24h period could recapitulate the effect of five days of stimulations (S1D-S1F Fig), these findings indicate the behavioural sensitization can be robustly induced across developmental stages. Next, we examined the impact of optogenetic stimulation intensity (applied between 96–120 h AEL) on nocifensive responses. Larvae exposed to higher LED intensities exhibited increased rolling probability, with behavioural effects reaching saturation at intensities above 0.35 µW/mm$^2$ (S1G-S1I Fig).

As noted above, previous studies have shown that continuous or high-frequency stimulation of nociceptive neurons can lead behavioural habituation [35,43,44]. To further investigate this in our paradigm, we varied the frequency of C4da neurons activation while maintaining stimulation duration and intensity (5s pulses and 5.6 µW/mm$^2$). As expected, high-frequency stimulation (every 15 or 30 seconds) significantly reduced rolling probability (20.8±6.1% and 35.2±6.8%, respectively), relative to naïve control (53.1 ± 5.9%). In contrast, lower-frequency stimulation (every 3, 5 or 15 minutes) induced significantly higher rolling probabilities (98.2 ± 1.8%, 98.9 ± 1.58%, 75.0 ± 8.5%, respectively) (Fig 1F), with an apparent transition point around a stimulation interval of once per minute (47.4 ± 7.0%). Rolling latency and total rolling duration exhibited similar trends (Fig 1G-1H). These results demonstrate that behavioural outcomes are highly sensitive to the temporal pattern of nociceptive input. They suggest that distinct and potential opposing mechanisms underlies the encoding of sensitization versus habituation evoked by prior experience.

## Behavioural sensitization reflects a decrease of the nociceptive threshold

The sensitization observed following optogenetic stimulation of C4da neurons during development could be an artifact of the technique rather than a biologically relevant process. If this was the case, optogenetically stimulated larvae might not show altered behavioural responses when presented natural noxious stimuli that also activate C4da neurons. To test this, we assessed the nocifensive response in both naïve and developmentally stimulated larvae following natural mechanical or chemical stimulation (Fig 2A). When tested with mechanical stimulation, developmentally stimulated larvae were significantly more likely to exhibit rolling behaviour than naïve controls (66% vs. 38%; Fig 2B). This increase in rolling was accompanied by reduction in the frequency of competing startle-like behaviour stopping (stimulated: 9% vs naïve: 25%) (Fig 2B). Similarly, following exposure to noxious chemical stimulus (5% hydrochloric acid, HCl), optogenetically stimulated larvae rolled more frequently than naïve controls (93% vs. 79%, as calculated by proportion of animals rolling within 10 seconds; Fig 2C). They also rolled with shorter latencies (3.27 ± 0.12 s vs 4.30 ± 0.15 s; Fig 2C).

These findings suggest that the increased rolling probability induced by optogenetic stimulation of C4da neurons may reflect a lowered threshold for nociceptive responses. If so, then dose-response curves for optogenetically stimulated

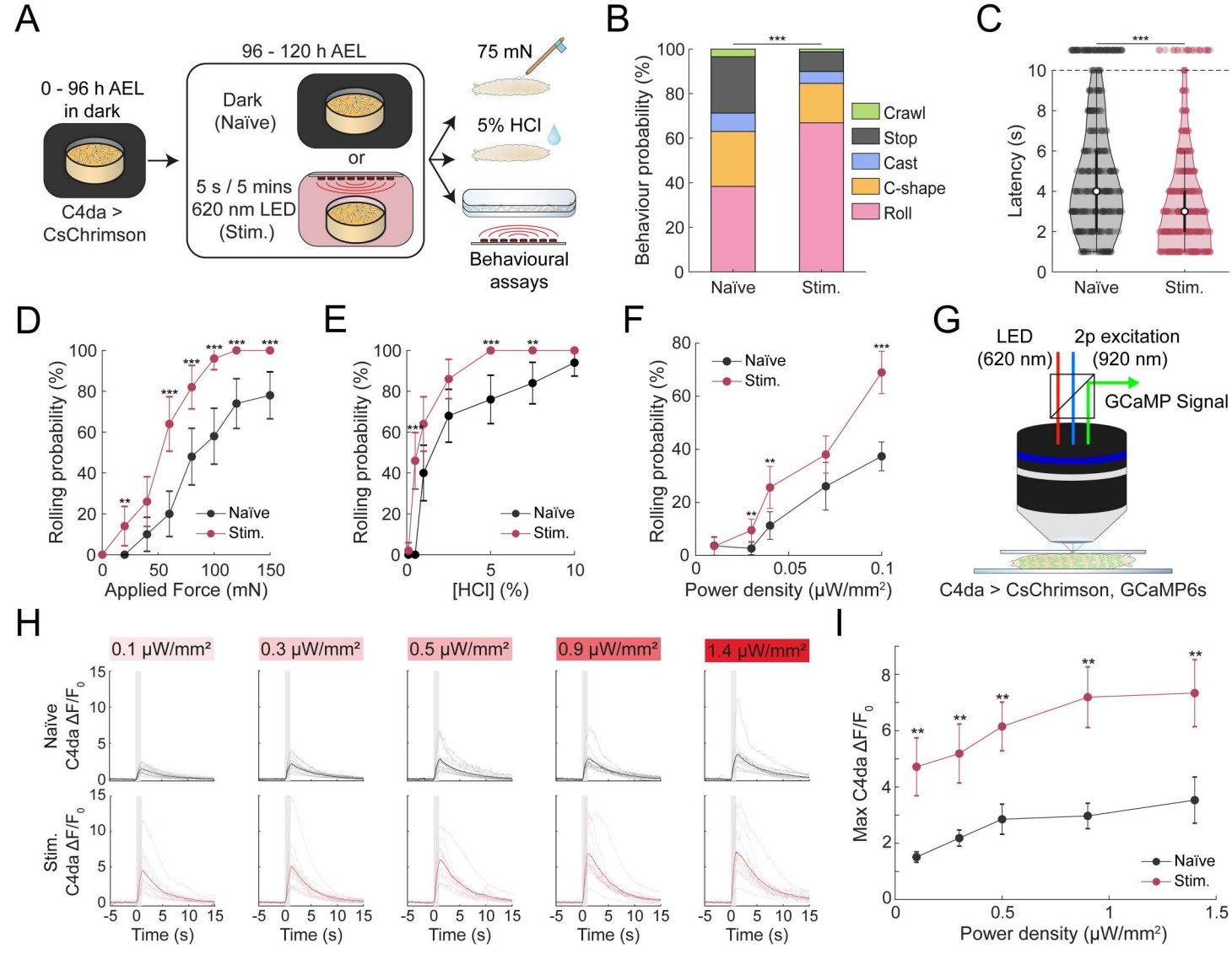

**Fig 2. Prior noxious experience enhances larval nociceptive responses via increased C4da neurons excitability sensitivity. (A)** Schematic of experimental design. Larvae were raised on all-trans retinal food, and developed in the dark until 96 hours post-collection. From 96h to 120h after collection, larvae were either kept in darkness (Naïve) or exposed to sporadic optogenetic activation of C4da neurons (Stim.). Nociceptive responses were then assessed using mechanical, chemical or optogenetic nociceptive assays. **(B)** Behavioural distribution in response to 75 mN mechanical stimulus, categorized into nociceptive responses (rolling and C-shape) and non-nociceptive (crawling, stopping, head-casting) responses (n = 146, 169). Mann-Withney U Test ***$p < 0.001$. **(C)** Violin plot of latency to roll following application of 5% HCl (1.5μL). Larvae not responding within 10 s were plotted above the cutoff (n = 343, 350). Mann-Whitney U Test, ***$p < 0.001$. **(D)** Dose-response curve for rolling probability as a function of force applied in the mechanical nociception assay. Error bars: 95% confidence interval (n = 50). Chi-square test, **$p < 0.01$, ***$p < 0.001$. **(E)** Dose-response curve for the rolling probability as a function of the concentration in the chemical nociception assay. Error bars: 95% confidence interval (n = 50). Chi-square test, **$p < 0.01$, ***$p < 0.001$. **(F)** Dose-response curve showing the rolling probability as a function of LED irradiance in optogenetic assays. Error bars: 95% confidence interval ($n_{Naïve}$ = 110, 189, 142, 92, 305; $n_{Stim}$ = 116, 189, 117, 184, 129). Chi-square test, ** $p < 0.01$, ***$p < 0.001$. **(G)** Schematics of in vivo 2-photon imaging setup for mechanically restrained 3rd instar larva. (H-I) Calcium imaging of GCaMP6s signal in C4da soma during optogenetic C4da stimulation. $n_{Naïve}$ = 12, 13, 12, 12, 11; $n_{Stim}$ = 10, 12, 10, 11, 10. **(H)** Individual and average ΔF/F$_0$ response traces at varying irradiances. Shaded regions mark stimulation period; thick traces indicated group means. **(I)** Maximum ΔF/F$_0$ across irradiances in naïve and stimulated animals. Error bars represent SEM. Mann-Whitney U Test ** $p < 0.01$.

versus naïve larvae should consistently show a leftward shift across different types of nociceptive stimuli. We first compared the mechanical thresholds between the two groups. At a typically subthreshold mechanical force (20 mN), 14 ± 9.6% of stimulated larvae exhibited rolling behaviour, whereas none of the naïve larvae responded. Stimulated larvae also showed significantly higher rolling probabilities at every points along the mechanical dose-response curve (40 mN:10 ± 8.3% vs 26 ± 12.2%; 60 mN: 20 ± 11.1% vs 64 ± 13.3%; 80 mN: 48 ± 13.8% vs 82 ± 10.6%; 100 mN: 58 ± 13.7% vs 96 ± 5.4%; 120 mN: 74 ± 12.2% vs 100%; 150 mN: 78 ± 11.5% vs 100%) (Fig 2D). We then tested chemical sensitivity using hydrochloric acid (HCl). Similar to mechanical stimulation, 46 ± 13.8% of the stimulated larvae rolled at concentrations that were subthreshold for naïve larvae (Fig 2E). Additionally, stimulated larvae displayed higher probabilities across nearly all concentrations (1% HCl: 40 ± 13.6% vs 64 ± 13.3%; 2.5% HCl: 68 ± 12.9% vs 86 ± 9.6%; 5% HCl: 76 ± 11.8% vs 100%; 7.5% HCl: 84 ± 10.2% vs 100%), except for the highest tested concentration (10% HCl: 94% vs 98%) (Fig 2E). Lastly, we assessed optogenetic stimulation. Developmentally stimulated larvae roll more frequently than naïve controls at the lowest light intensity ($0.03 \, \mu W/mm^2$ – 2.65 ± 2.28% vs 9.52 ± 4.18%) and showed increased rolling probabilities at all higher intensities ($0.04 \, \mu W/mm^2$: 11.3 ± 5.20% vs 25.6 ± 7.91%; $0.07 \, \mu W/mm^2$: 26.1 ± 8.97% vs 38.0 ± 7.01%; $0.1 \, \mu W/mm^2$: 37.4 ± 5.43% vs 69.0 ± 7.98%) (Fig 2F).

Collectively, these data support the conclusion that developmental optogenetic stimulation of the C4da neurons induces behavioural sensitization by lowering the threshold for nociceptive responses across mechanical, chemical and optogenetic stimuli.

## Previous noxious experiences alter C4da neuron responses to stimuli, but not morphology

To understand the neural mechanisms underlying sensitization, we examined how developmental stimulation affect the activity of C4da sensory neurons. Larvae expressing both the calcium indicator GCaMP6s and optogenetic actuator CsChrimson in C4da neurons were subjected to acute optogenetic stimulation, and calcium responses were imaged in the dendritic field near the soma (Fig 2G). Compared to naïve controls, developmentally stimulated larvae exhibited significantly higher calcium responses across all tested light intensities ($\Delta F/F_0$; naïve vs stim: $0.1 \, \mu W/mm^2$: 1.51 ± 0.19 vs 4.72 ± 1.03; $0.3 \, \mu W/mm^2$: 2.18 ± 0.29 vs 5.19 ± 1.05; $0.5 \, \mu W/mm^2$: 2.85 ± 0.53 vs 6.15 ± 0.86; $0.9 \, \mu W/mm^2$: 2.97 ± 0.45 vs 7.19 ± 1.08; $1.4 \, \mu W/mm^2$: 3.53 ± 0.82 vs 7.34 ± 1.19) (Fig 2H-2I and S1-S2 Movies). These results indicate that C4da neurons become more responsive following noxious stimulation during development.

Given the change in dendritic arborization are often associated with altered sensory neuron sensitivity [45,55–57], we next examined whether morphological changes might account for the increased activity. To this end, a Sholl analysis was performed on C4da dendritic arbors in developmentally stimulated (5 s/ 5 min., $5.6 \, \mu W/mm^2$ for 96–120 AEL) and naïve larvae. This revealed that stimulation of C4da neurons during larval development did not drive changes in their arborization (S2 Fig). While the Sholl profile suggested a mild increase in branching in stimulated animals, these differences were not statistically significant (S2A–S2B Fig). There were no significant differences in key parameters including the area under the Sholl curve (naïve vs stim.: 23.0 ± 1.24 vs 25.7 ± 1.32), critical radius (50.1 ± 3.4% vs 52.2 ± 4.1%), or number of intersections at critical radius (47.8 ± 2.8 vs 52.6 ± 3.6) (S2C-S2E Fig). These findings indicate that developmental sensitization of C4da neurons is not mediated by structural change of their dendritic arbors. Instead, sensitization likely arises from functional changes driven by prior neuronal activity.

## The OAMB receptor in C4da neurons is necessary for experience-dependent sensitization

A previous study reporting behavioural habituation following continuous noxious chemical stimulation of C4da neurons identified serotonin as a key neuromodulator mediating this effect [43]. To investigate whether neuromodulators also play a role in the experience dependent sensitization observed in our experimental paradigm, we performed RNAi knockdown of various neuromodulator receptors specifically in C4da neurons. The behavioural responses of both naïve and developmentally stimulated larvae with and without receptor knockdown was compared.

Although it is unclear whether the RNAi knockdown was inefficient, this screen identified the octopamine receptor OAMB (octopamine receptor in mushroom bodies) as a strong candidate mediating the sensitization of C4da neurons (S3A–S3C Fig and 3A Fig). RNAi knockdown of OAMB using two independent lines had minimal effect on nocifensive behaviour in naïve larvae, with only slight reduction or no reduction in rolling probability, rolling latency or total rolling duration compared to control naïve animals (Fig 3A–3C), despite both RNAis being able to successfully downregulate OAMB expression

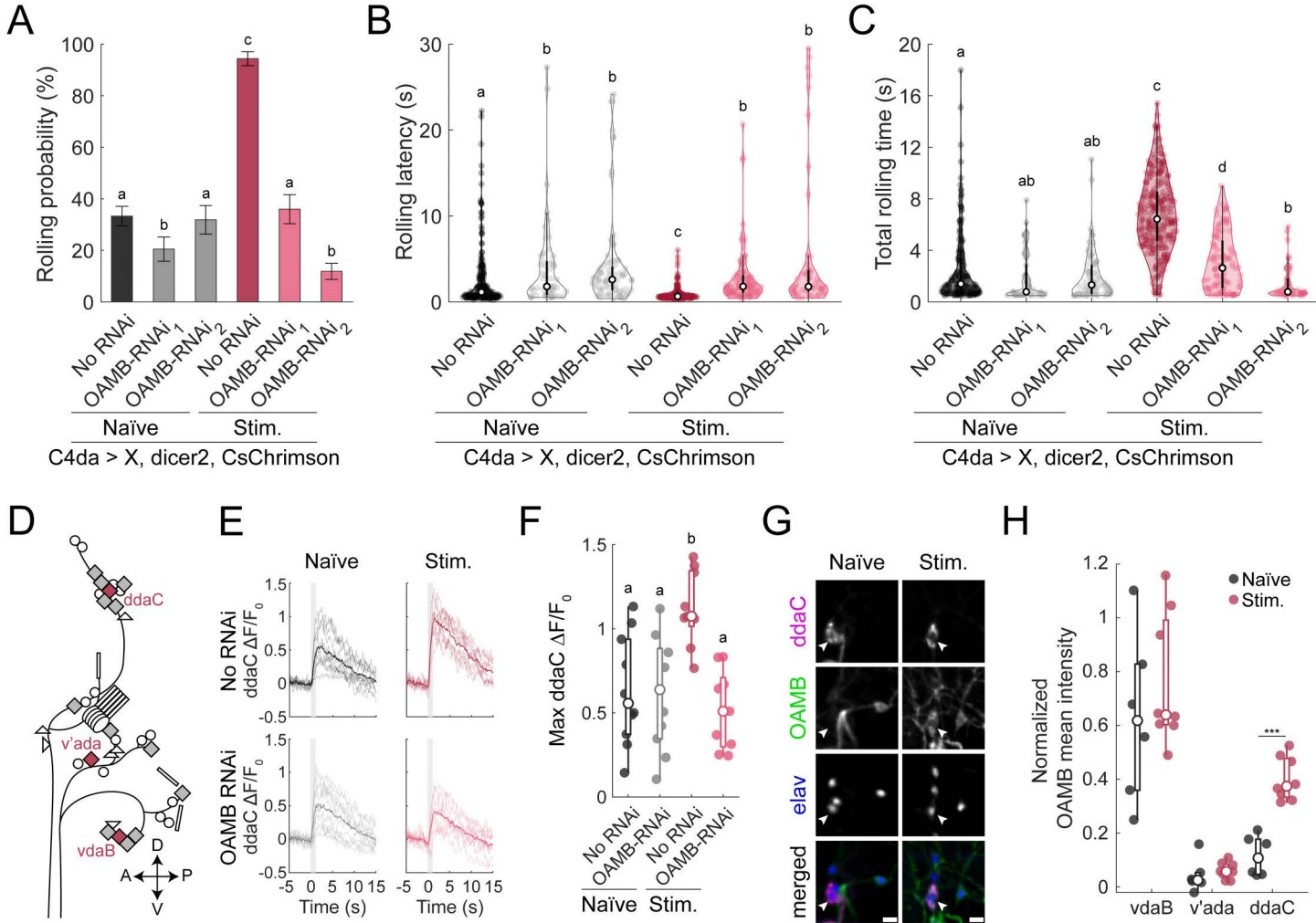

**Fig 3. The octopamine receptor OAMB is required for experience-dependent sensitization to noxious stimuli. (A-C)** RNAi knockdown of OAMB blocks the behavioural sensitization induced by prior noxious experience. **(A)** Rolling probability during optogenetic C4da stimulation. Error bars: 95% confidence interval (n = 589, 283, 270, 284, 278, 414). Chi-square test; compact letter display (CLD) shows statistical significance at $p < 0.01$. **(B)** Violin plot of rolling latency during optogenetic stimulation (n = 195, 58, 86, 268, 100, 49). Kruskall-Wallis followed by post-hoc Mann-Whitney test, CLD denotes $p < 0.01$. **(C)** Violin plot of total time each larvae spent rolling during optogenetic stimulations (n = 195, 58, 86, 268, 100, 49). Welch's ANOVA with post-hoc Mann-Whitney test; CLD denotes $p < 0.01$. **(D)** Schematic of the larval somatosensory system. Triangles and circles represent external sensory neurons, rectangles represent the ciliated chordotonal neurons, and rotated-square represent multidendritic neurons. The C4da neurons (vdaB, v'ada, ddaC) are highlighted in red. **(E-F)** Calcium imaging of ddaC neuronal responses during optogenetic stimulation with or without OAMB RNAi. n = 10, 10, 9, 9. **(E)** $\Delta F/F_0$ traces from individual neurons aligned to stimulus onset. Shaded region indicates stimulation period. **(F)** Maximum $\Delta F/F_0$ responses in naïve and stimulated larvae with or without RNAi. Mann-Whitney test; CLD indicates statistical significance at $p < 0.01$. **(G-H)** OAMB expression varies with neuronal identity and prior noxious experience. **(G)** Representative images mean OAMB expression in ddaC neurons in naïve and stimulated larvae. Scale bar = 10 μm. **(H)** Boxplot of normalized OAMB expression (relative to elav staining) across C4da neuron types and groups. (n_naïve = 6, n_stim = 8). One-way ANOVA Test ***$p < 0.001$.

(S3D Fig). However, in developmentally stimulated larvae, OAMB knockdown abolished the enhanced rolling response. Specifically, rolling probability was significantly reduced in OAMB knockdown animals compared to stimulated controls (No RNAi: 94.6 ± 2.64%; OAMB-RNAi-1: 36.0 ± 5.64%; OAMB-RNAi-2: 11.8 ± 3.11%) (Fig 3A). In addition, latency to roll was increased (No RNAi: 0.87 ± 0.04 s; OAMB-RNAi-1: 2.85 ± 0.33 s; OAMB-RNAi-2: 5.50 ± 1.12 s), and the total rolling time was decreased (No RNAi: 6.76 ± 0.18 s; OAMB-RNAi-1: 3.05 ± 0.22 s; OAMB-RNAi-2: 1.43 ± 0.19 s) (Fig 3B–3C). These results demonstrate that OAMB is necessary in C4da neurons for experience-dependent sensitization prevents induced by developmental nociceptive stimulation. Knockdown of this octopamine receptor effectively blocks the increased behavioural sensitivity, suggesting that octopaminergic signaling acts directly on C4da neurons to mediate sensitization.

We next examined whether RNAi-mediated knockdown of OAMB affected the activity of C4da neurons *in vivo* by imaging calcium dynamics. It must be noted that C4da neurons is a class of nociceptive sensory neurons, which comprises the ventral vdaB, the lateral v'ada and the dorsal ddaC (Fig 3D). Thus, we specifically examined the calcium dynamics of ddaC neuron. In naïve larvae, GCaMP signals in response to optogenetic activation of ddaC neurons were unaffected by OAMB knockdown (RNAi: 0.62 ± 0.11, no RNAi: 0.63 ± 0.10) (Fig 3E-3F). However, in developmentally stimulated larvae, GCaMP signals were significantly elevated in ddaC neurons with intact OAMB expression (1.12 ± 0.07), while this activity enhancement was abolished by OAMB knockdown (0.52 ± 0.08) (Fig 3E-3F). A similar pattern was observed when imaging the axon terminal of all C4da neurons (S3E-S3F Fig). These results suggest that OAMB is required for the experience-dependent increase in sensory neuron activity following developmental stimulation.

To our knowledge, no prior studies have demonstrated OAMB expression in C4da neurons. To address this, we used MiMIC-converted Trojan-GAL4 line targeting an intronic region shared by all OAMB isoforms [21], to assess endogenous OAMB expression. We quantified co-expression of MiMIC-driven GFP with tdTomato-labeled C4da neurons proxy for OAMB expression in naïve and experienced larvae. GFP signal from OAMB-trojan-GAL4 was consistently high in vdaB neurons, independent of experience (naïve: 0.63±0.13, experience: 0.76±0.09), but not higher than background levels in ventral v'ada neurons (naïve: 0.04±0.03, experienced: 0.06±0.01) (Figs 3H and S3G-S3H). In contrast, ddaC neurons exhibited negligible OAMB-trojan-GAL4 signals in naïve larvae (0.12±0.03), but significantly elevated levels following developmental stimulation (0.40±0.03) (Fig 3G-3H). These data indicate that while some C4da neurons (e.g. vdaB) show consistently high OAMB-trojan-GAL4 signals, ddaC neurons exhibit experience-dependent increase in reporter signal. This suggests that plastic changes in OAMB expression levels may serve as mechanism for observed sensitization.

To support this idea, we examined whether OAMB overexpression in C4da neurons was sufficient to induce sensitization. It must be noted that *Drosophila* has two distinct functional OAMB isoform, OAMB-AS and OAMB-K3. Studies conducted in vitro indicates that both isoforms can increase calcium concentration, but K3 isoform can additionally increase cAMP levels [58,59]. Thus, two distinct effectors that encoded wild-type version of either isoform were used [60]. Surprisingly, overexpression of OAMB-AS, but not OAMB-K3, increases rolling probability and decreases rolling latency (S3I-S3J Fig). Further, OAMB-AS overexpression significantly increase total rolling time, but OAMB-K3 overexpression only trends towards such an increase (p=0.0824) (S3K Fig). While the reason behind this behavioural difference is unclear, these results suggest that increase in OAMB expression could have a causal role in the sensitization of C4da neurons through the receptor's ability to increase intracellular calcium concentrations. However, it must be noted that even OAMB-AS overexpression across development is not sufficient to fully replicate the effect of experience, with weaker effects on latency and total rolling time when comparing to 24h stimulations (Cohen's d, experience vs OE; latency: 0.737 vs 0.558; total rolling time: 1.655 vs 0.753) (Figs 3B-3C and S3I-S3K). This data suggests that while OAMB-AS isoform might be a key receptor, it is insufficient on its own to fully account for the behavioral sensitization evoked by nociceptive sensory activation.

## Octopaminergic signalling by tdc2+neurons during noxious experience is necessary for nociceptive sensitization

Since OAMB in C4da neurons is required for experience-dependent sensitization and that changes in OAMB expression only partially replicates the effect of experience, we surmised that the ligand of this receptor, octopamine, plays

an essential role. To investigate how octopaminergic neurons contribute to the encoding of prior noxious experiences within the nociceptive system. Tdc2-GAL4 line was used to drive expression of two RNAi lines targeting tyramine beta-hydroxylase (tbh-RNAi-1: JF02746, tbh-RNAi-2: HMS05829), a rate-limiting enzyme in octopamine synthesis. Tdc2-GAL4 driver recapitulates the expression pattern of tyrosine decarboxylase 2 gene and has been shown to drive expression in both tyraminergic and octopaminergic neurons [61,62]. Consequently, this driver was previously used to study the role of octopaminergic neurons in larvae [61,62].

Naïve larvae expressing tbh-RNAi-1 or tbh-RNAi-2 in C4da neurons displayed nocifensive behaviour comparable to those of naïve controls, although each RNAi background led to a difference in baseline behaviour. (Fig 4A and S4A-S4B Fig). As both RNAi are efficient in reducing tbh expression in neural tissues (S4C Fig), this lack of effect in naïve larvae points towards octopamine not being critical to normal nociception. In contrast, stimulated larvae in which production of octopamine was knocked down by tbh-RNAi-1 or tbh-RNAi-2 showed reduced sensitization compared to stimulated controls, as assessed by the rolling probability (tbh-RNAi-1: 65.9 ± 5.02% vs control: 83.2 ± 3.91%; tbh-RNAi-2: 58.6 ± 5.72% vs control: 77.7 ± 5.61%), total rolling time (tbh-RNAi-1: 5.16 ± 0.31 s vs control: 8.52 ± 0.42 s; tbh-RNAi-2: 11.4 ± 0.56 s vs. control: 16.4 ± 0.55 s), and rolling latency (tbh-RNAi-1: 9.30 ± 0.35 s vs. control: 5.23 ± 0.26 s; tbh-RNAi-2: 3.29 ± 0.27 s vs control 0.84 ± 0.18 s) (Fig 4A and S4A-S4B Fig). These results support a model in which octopaminergic signalling from tdc2+ neurons contributes to the sensitization of C4da neurons.

To further test the role of tdc2+ neurons in sensitization, we devised an experiment to transiently inhibit the activity of tdc2 + neurons during noxious stimulation. We expressed green-light gated anion channel GtACR1 in tdc2+ neurons, while simultaneously expressing CsChrimson in C4da neurons. Larvae were then subjected to developmental stimulation using 525-nm (green) light (Fig 4B). This wavelength effectively activates both GtACR1 and CsChrimson [63,64]. Thus, a 525-nm light should supress tdc2+ neurons activity during optogenetic activation of C4da neurons. In contrast, stimulation with 620-nm (red) light should activate C4da neurons without affecting GtACR1 in tdc2+ neurons [63,64]. While the presence of tdc2-GAL4 lead to substantially heightened rolling probability, GtACR1 expression in tdc2+ neurons did not alter the

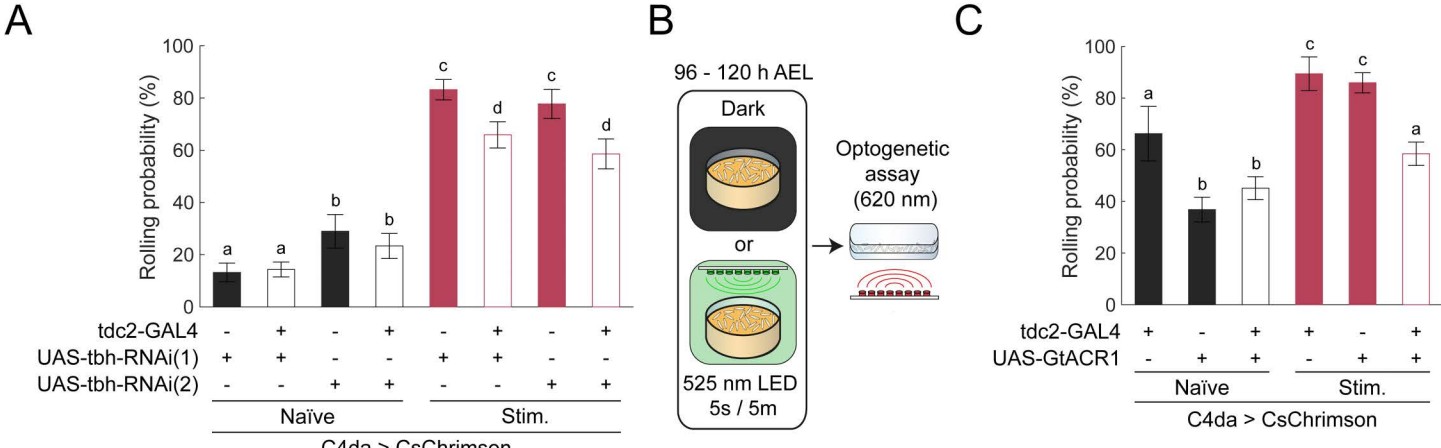

**Fig 4. Octopaminergic signalling from tdc2 + neurons is required for experience-dependent nociceptive sensitization. (A)** RNAi knockdown of tbh in tdc2+ neurons impairs experience-dependent sensitization without affecting baseline nociceptive responses. Rolling probability during optogenetic stimulation (n = 334, 579, 190, 300, 351, 343, 211, 285). Error bars represent 95% confidence interval. Chi-square test; CLD indicates statistical significance (p < 0.01). **(B)** Schematic of experimental design for inhibition of tdc2+ neurons during C4da neurons stimulation. 525 nm LED activate GtACR1 in tdc2+ neurons and CsChrimson in C4da neurons during development. Behavioural test was performed using 620 nm LED that only activate CsChrimson in C4da neurons. **(C)** Silencing tdc + 2 neurons using GtACR1 alters nociceptive behaviour only in developmentally stimulated larvae. Rolling probability during optogenetic C4da neurons. (n = 77, 391, 483, 85, 306, 455). Error bars represent 95% confidence interval. Chi-square test; CLD indicates statistical significance (p < 0.01).

probability of rolling in naïve larvae compared to UAS-GtACR1 background (tdc2 > -: 66.23 ±10.566%, - > GtACR1: 36.8 ± 4.8%, tdc2 > GtACR1: 45.1 ± 4.4%) (Fig 4C). When tdc2+ neuron activity was inhibited during C4da neuron activation, there was a significant reduction, but not a complete elimination, of sensitization in developmentally stimulated larvae (tdc2 > −: 89.41±6.54%, −>GtACR1 control: 86.0±3.89%, tdc2 > GtACR1: 58.5±4.5%) (Fig 4C). This indicates that other neurons may also contribute to the sensitization phenotype.

Since the activation of C4da during development results in both decreased response latency and increased total rolling time, we next assessed whether inhibition of tdc2+ neurons affect these parameters. Inhibiting tdc2+ neurons activity partially attenuated the reduction in rolling latency following stimulation (tdc2 >-, naïve vs stim.: 3.02 ± 0.36s vs 2.12 ± 0.32 s;->GtACR1, naïve vs stim.: 4.24 ± 0.45 s vs 1.68 ± 0.15 s; tdc2 > GtACR1, naïve vs stim.: 2.73 ± 0.36 s vs 1.25 ± 0.19 s – Cohen's d = 0.455 vs 0.760 vs 0.343). However, inhibition of tdc2+ neurons did not prevent prolongation of total rolling time (tdc2 >-, naïve vs stim.: 3.96 ± 0.47 s vs 6.99 ± 0.53 s; - > GtACR1, naïve vs stim.: 3.40 ± 0.3 s vs 14.3 ± 0.5 s; tdc2 > GtACR1, naïve vs stim.: 5.78 ± 0.34 s vs 18.6 ± 0.50 s), (S4C-S4D Fig). Together, these finding demonstrate that octopaminergic signalling during noxious stimulation is critical for experience-dependent sensitization, consistent with the results observed using tbh-RNAi. Specifically, activity of tdc2+ neurons during harmful experience is required for the increase in rolling probability and the reduction in response latency, but not for the prolongation of the overall nocifensive behaviour.

## Octopamine is sufficient to induce sensitization of the nociceptive system

Having shown that octopaminergic activity is necessary for sensitization, we next investigated whether octopamine itself is sufficient to induce sensitization in larval nociception. To test whether octopamine could mimic effect of nociceptive stimulation during development, larvae were given food supplemented with octopamine at concentrations previously shown to rescue phenotypes in octopamine-deficient larvae [18,65]. In line with prior studies [18,65], and to minimize potential acclimation to the additive, larvae were fed octopamine-enriched food for two hours prior to nociceptive optogenetic, mechanical, or chemical assay (Fig 5A). When optogenetically activating C4da neurons, dietary octopamine significantly increased the rolling probability at all tested concentrations (5 mM: 69.1 ± 6.15%, 25 mM: 72.3 ± 6.07%, 50 mM: 77.0 ± 6.25%) compared to untreated control (0 mM: 49.0 ± 7.03%) (Fig 5B). Since 5 mM was sufficient to elicit behavioural changes, this octopamine concentration was used for subsequent experiments. In the mechanical stimulation assay, larvae exposed octopamine exhibited increased rolling probability (52%) compared to untreated control (31%) (Fig 5C). Similarly, in the chemical stimulation assay, octopamine-fed larvae showed an elevated (98%) rolling probability relative to control (92%) and reduced rolling latency (control: 3.16 ± 0.18 s; 5 mM 2.53 ± 0.14 s) (Fig 5D).

These data support the idea that octopamine enhances larval responses to noxious stimuli. This effect could result from changes in sensory neurons sensitivity, or from alterations in motor neurons where octopamine is known to regulates locomotion by modulating motor neuron activity through a balance between tyramine and octopamine [18,23,66]. To determine whether the behavioural effects of octopamine were at least partially derived from changes in sensory processing, we examined the calcium responses in C4da neurons in whole-mount larvae following octopamine treatment. Consistent with responses observed in larvae previously exposed to noxious stimulation, octopamine-fed larvae exhibited significantly enhanced calcium activity in C4da neurons upon optogenetic stimulation (Fig 5E-5F). Collectively, these results suggest that the octopamine sensitizes C4da neurons that likely underlies the observed behavioural changes.

We next conducted OAMB RNAi knockdown paired with octopamine feeding to test whether the effect of octopamine on nociception is dependemt on the basal OAMB receptor expression in C4da neurons. Here, OAMB RNAi knockdown in unfed larvae did not decrease rolling probability, with one RNAi line instead producing a modest but significant increase in rolling behaviour (No RNAi: 30.65 ± 6.41%; OAMB-RNAi-1: 30.84 ± 6.01%; OAMB-RNAi-2: 46.05 ± 7.92%) (Fig 5G). In contrast, RNAi knockdown of OAMB significantly dampened the effect of octopamine feeding, resulting in a significant reduction of the rolling probability (No RNAi: 52.94 ± 7.15%; OAMB-RNAi-1: 39.8 ± 6.77%; OAMB-RNAi-2: 32.99 ±

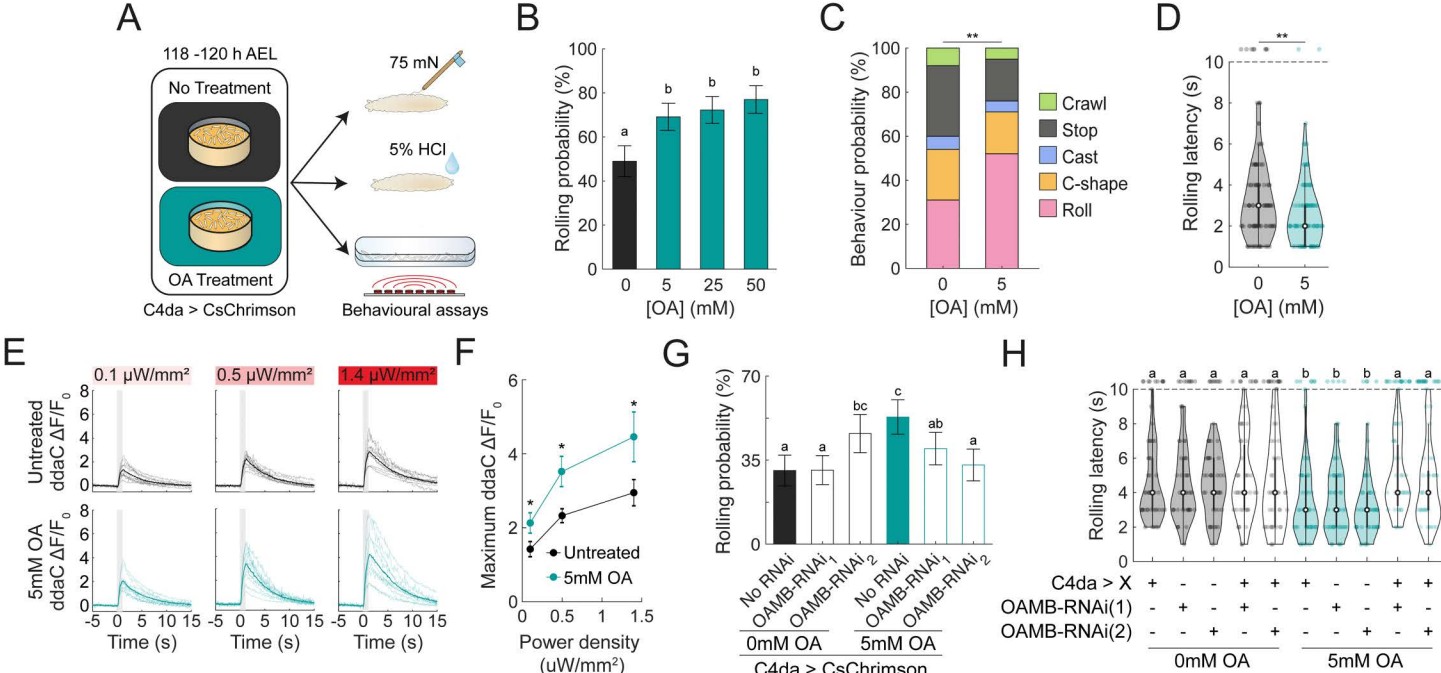

**Fig 5. Octopamine is sufficient to enhance nociceptive response. (A)** Schematics of experiment. Larvae were fed food supplemented with octopamine at varying concentrations prior to behavioural assays. **(B-D)** Octopamine feeding increases rolling in response to real and fictive nociceptive stimuli. **(B)** Rolling probability over the 30 s optogenetic stimulation of C4da neurons. Bars indicate mean; error bars denote 95% confidence intervals (n = 194, 217, 209, 174). Chi-square test; CLD indicates statistical significance ($p < 0.01$). **(C)** Behavioural distribution in response to 75 mN mechanical stimulus (n = 100/group). Mann-Whitney U Test, **$p < 0.01$. **(D)** Rolling latency following 5% HCl exposure (n = 100/group). Mann-Whitney U Test, **$p < 0.01$. **(E-F)** Octopamine feeding enhances C4da neuronal responses to optogenetic C4da stimulation. **(E)** Calcium traces from individual C4da neurons. Shaded regions denote stimulus period: bold line show group means. ($n_{Untreated}$ = 10, 10, 10; $n_{Treated}$ = 10, 12, 10) **(F)** Maximum ΔF/F0 in untreated and treated larvae. Mann-Whitney U Test *$p < 0.05$. **(G-H)** RNAi knockdown of OAMB in C4da neurons prevents the effect of octopamine feeding on nociceptive behaviour. **(G)** Rolling probability over the 30 s optogenetic stimulation of C4da neurons. Bars indicate mean; error bars denote 95% confidence intervals (n = 199, 227, 152, 187, 201, 194). Chi-square test; CLD indicates statistical significance ($p < 0.01$). **(H)** Rolling latency following 5% HCl exposure (n = 50/group). Kruskall-Wallis test followed by post-hoc pairwise Mann-Whitney test; CLD indicates statistical significance at $p < 0.01$.

6.62%) (Fig 5G). To confirm that these effects were not due to background effects inherent to our optogenetic paradigm, larvae were assessed in a chemical nociception assay. In unfed larvae, there was no significant differences between RNAi knockdown and their respective genetic controls (Fig 5H). However, in octopamine-fed larvae, the RNAi knockdown prevented the effects of octopamine, with larvae expressing RNAis displaying lower rolling latency and probability than octopamine-fed controls (Fig 5H). Taken together, these data suggest that modulation of C4da neurons activity can be achieved through an increase in octopaminergic signal, which can be detected through basal OAMB expression.

## Ventral unpaired median neurons (VUMs) are critical for experience dependent sensitization

In *Drosophila* larvae, approximately 100 octopaminergic neurons are present in the central nervous system (CNS) [67,68]. Octopaminergic neurons in the ventral nerve cord (VNC) can be classified into distinct types, including ventral unpaired median neurons (VUMs), a class of segmentally repeated type II motor neurons; thoracic and posterior (A9) dorsal unpaired median neurons (DUMs); abdominal leucokinin-producing neurons (ABLKs); and thoracic and anterior abdominal (A1/A2) ventral paired median neurons (VPMs) [67–69]. Although prior connectome studies indicate that VUM neurons are immature in first-instar larvae [70], to investigate the anatomical relationship between octopaminergic neurons and C4da sensory neurons in third-instar larvae, we first employed immunohistochemistry to compare the morphology

of tdc2$^+$ neurons and C4da neurons, carefully tracking overlapping processes to infer neuronal identity. This analysis revealed that VUM neurons extend processes that overlap with both anterior and posterior regions of C4da axon terminals (Fig 6A-6B).

To further test the role of VUMs in experience-dependent sensitization, we used targeted genetic approaches to manipulate CsChrimson expression. Specifically, tsh-LexA drove FLP or GAL80, restricting CsChrimson expression to either brain or the VNC (S5A-S5C Fig). While ABLKs were not found in any of stainings for tdc2, this intervention was successful in segregating neuronal populations from one another (S5A-S5C Fig). In parallel, we used VUM-GAL4 line (GMR46B08-GAL4), which selectively targets expression of CsChrimson to VUM neurons and small subset of secondary abdominal neurons (S5D Fig). Larvae expressing CsChrimson in the targeted neuronal populations for 24 hours (5 seconds every 5 minutes), and behavioural responses were compared the results to control groups: larvae lacking GAL4 (no CsChrimson expression) and larvae expressing CsChrimson in all tdc2 + neurons. We hypothesized that if the tdc2 + neurons activity is critical for nociceptive sensitization, their stimulation would at least partially mimic the effect of C4da neurons stimulation. Moreover, if VUM neurons are the primary mediators of this effect, then stimulating them in isolation should be sufficient drive similar behavioural sensitization. Supporting this hypothesis, optogenetic activation of tdc2 + neurons in VNC neurons alone or of the VUM in isolation significantly increased the rolling probability in response to a mechanical stimulus, relative to naïve controls or larvae lacking CsChrimson expression (Fig 6C). A similar effect was observed in response to noxious chemicals, with treated larvae displaying both reduced rolling latency and increased rolling probability (Fig 6D). These effects were comparable to those observed in larvae when all tdc2 + neurons were activated (Fig 6C-6D). In contrast, activation of tdc2 + neurons in the brain did not significantly alter nociceptive responses (Fig 6C-6D). Taken together, these findings strongly suggest that activation of VUM neurons activation is sufficient to drive experience-dependent sensitization of nociceptive behaviour. This likely occurs through octopaminergic modulation of C4da neurons.

Having established that VUMs activity is critical for nociceptive sensitization, we next investigated the functional relationship between VUMs and C4da neurons, both in naïve and experienced larvae. To assess potential anatomical connectivity, we employed activity-dependent GFP reconstitution across synaptic partner (syb-GRASP) [71], assuming that C4da neurons function as presynaptic partner to VUMs. In line with our anatomical descriptions (Fig 6A-6B), naïve animals displayed sparse but consistent GRASP signals, most reliably in segments T2, A2-A4 and A8/9 (Fig 6E and S1 Table). To further evaluate functional connectivity, we imaged calcium responses in VUM neurons following optogenetic activation of C4da neurons. While naïve larvae exhibited moderate calcium transients (average max ΔF/F0: 0.76 ± 0.13), experienced larvae displayed significantly larger responses (2.67 ± 0.36), indicating that prior noxious stimulation enhances functional connectivity from C4da neurons to VUMs (Fig 6F-6G).

To understand if the functional changes between C4da neurons and VUMs was relevant to plasticity in nociceptive behaviour at large, we examined the activity of VUMs under habituated conditions. When stimulated at a high frequency, VUMs' peak activity is not significantly altered relative to naïve controls (S6A-S6B Fig). However, VUM are slower to reach that peak, suggesting that high frequency stimulation affects the temporal dynamics of these neurons (S6C Fig). To understand if this is an inherent property (i.e., frequent stimulations delays future activation of octopaminergic neurons), we proceeded to stimulate tdc2 neurons at a high frequency, and to contrast with high frequency stimulation of C4da neurons in a chemical nociception assay. While high frequency stimulation of C4da neurons drastically decreased responsivity to noxious chemicals compared to all controls, stimulations of tdc2$^+$ neurons had the opposite effect (S6D Fig). Taken together, these results suggest that the change in VUMs' activity under a habituation paradigm is not due to an inherent property of these neurons, with habituation likely relying on distinct mechanisms.

To explore whether the interaction between VUMs and C4da neurons is unidirectional or bidirectional under a sensitization paradigm, we activated tdc2 + neurons using CsChrimson and monitored octopaminergic transmission in C4da neurons using octopamine sensor GRAB-OA1.0 [15]. This analysis revealed that C4da axon terminals receive octopaminergic input upon activation of tdc2 + neurons, and this input is significantly elevated in larvae with prior activation of the

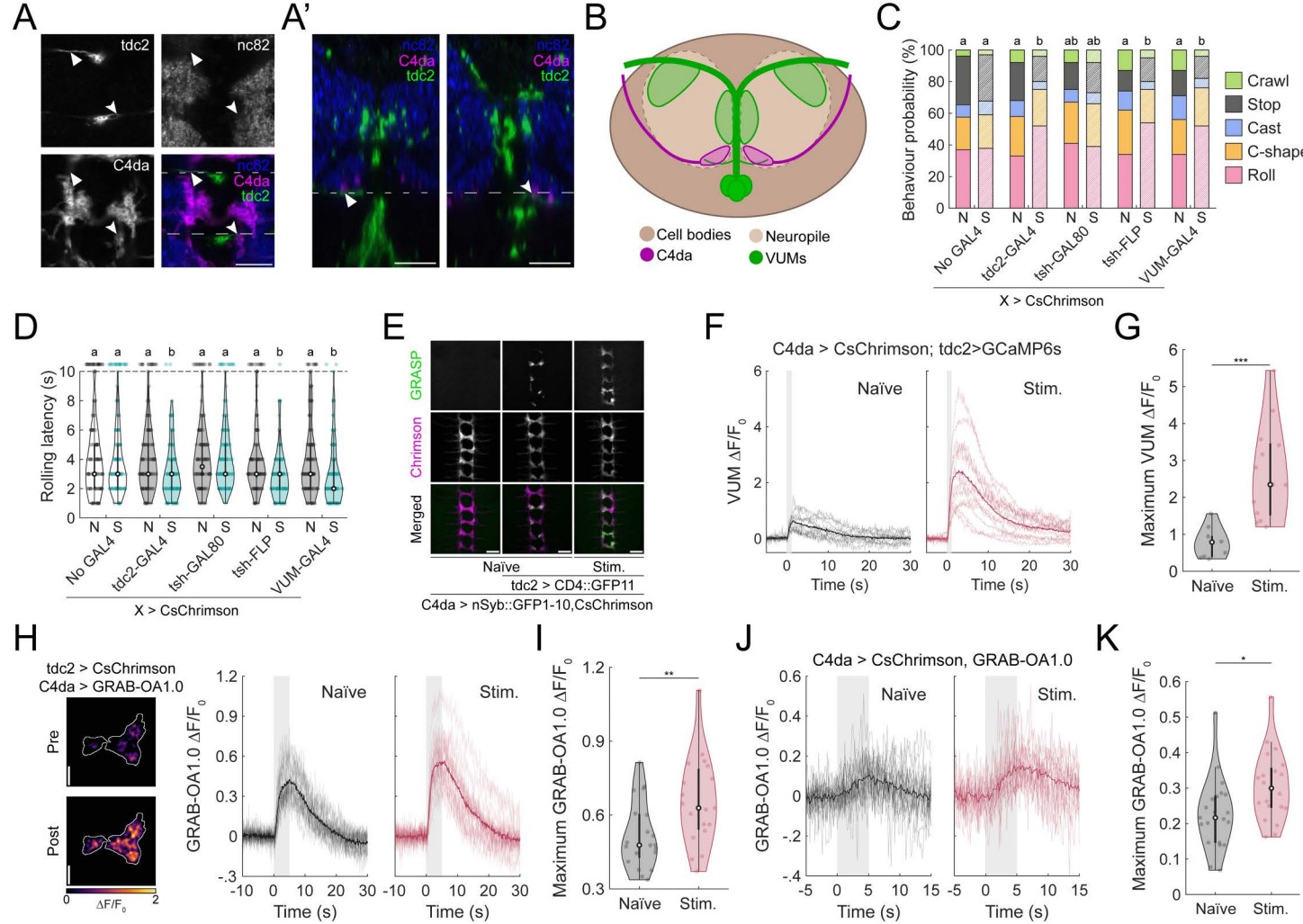

**Fig 6. Octopaminergic VUM neurons mediate experience-dependent sensitization in the nociceptive circuit.** (A-A') Anatomical relationship between tdc2+ neurons and C4da neurons at abdominal segment A4. Tdc2-LexA was used to drive expression of mCD8::GFP (green) in tdc2+ neurons, while ppk1.9-GAL4 drove mCD8::RFP (magenta), tagging the cell membrane of both neuron population. nc82 (blue) is used as a counterstaining for the neuropil and indicates active zones within the brain. **(A)** Dorsal view. (A') Transverse view. Both views: Arrow heads and triangle indicate points of contact between the abdominal VUM arbor and C4da axon terminal and point at the same coordinates. Dashed lines indicate the position at which views intersect. Scale bar = 20 μm. **(B)** Diagram summarizing the anatomical relationship between VUM neurons and C4da neurons. VUM neurons extent a lateral process that intersects with the anterior/posterior region of the C4da neuron axon terminal. **(C-D)** Stimulation of VNC-restricted tdc2+ neurons recapitulate the sensitization induced by full tdc2+ neuron activation. N denotes naïve larvae, S indicates previously stimulated larvae. **(C)** Behavioural distribution in response to 75 mN mechanical stimulus (n = 205, 100, 100, 100, 100, 198, 100, 100, 100, 100). Kruskall-Wallis test followed by post-hoc pairwise Mann-Whitney test; CLD indicates statistical significance at $p < 0.01$. **(D)** Violin plot of latency to roll following application of 5% HCl (n = 113, 100, 100, 100, 100, 112, 100, 100, 100, 100). Kruskall-Wallis test followed by post-hoc pairwise Mann-Whitney test; CLD indicates statistical significance at $p < 0.01$. **(E)** C4da neurons forms direct synapses to tdc2+ neurons. Representative image of trans-synaptic GRASP signal between C4da and tdc2+ neurons between segments A1 and A5. Scale bar = 20 μm. **(F-G)** Calcium imaging of tdc2+ VUM neurons reveals enhanced activation by C4da input after noxious developmental stimulation. **(F)** GCaMP6s traces of individual and average response in VUM neurons; shaded region indicates stimulus. **(G)** Peak $\Delta F/F_0$ in naïve vs. stimulated larvae (n = 10, 13). Mann-Whitney test *** $p < 0.001$. **(H-I)** GRAB-OA1 imaging of C4da neurons in response to tdc2+ neurons simulation. **(H)** Heatmap represent average intensity of GRAB-OA signals in C4da axon terminal before and after stimulation in a naïve animal (left panel). GRAB-OA traces of individual and average C4da neurons; shaded region indicates stimulus. **(I)** Maximum $\Delta F/F_0$ in naïve and stimulated larvae (n = 20, 19). One-way ANOVA, ** $p < 0.01$. **(J-K)** GRAB-OA1 imaging of C4da neurons in response to C4da neurons simulation. **(J)** GRAB-OA traces of individual and average C4da neurons; shaded region indicates stimulus. **(K)** Maximum $\Delta F/F_0$ in naïve and stimulated larvae (n = 21, 20). One-way ANOVA, * $p < 0.05$.

tdc2 + neurons (Fig 6H-6I). Finally, to validate the relevance of this mechanism, we examined the feedback received by C4da axon terminal upon activation of C4da neurons in naïve and previously stimulated larvae. Here, the results indicates that previous noxious experience heightens the octopaminergic signal received by C4da neurons upon subsequent reactivation (Fig 6J-6K). Together, these findings support the existence of positive feedback loop between C4da sensory neurons and VUMs octopaminergic neurons that allows for sensitization. In this circuit, noxious stimulation strengthens the functional connectivity and mutual responsiveness: C4da neurons more effectively activate VUMs, while VUMs, in turn release greater amounts of octopamine onto C4da neurons in experienced larva, which is detected by OAMB (Fig 7). This bidirectional enhancement provides to amplify nociceptive sensitization following developmental experience.

## Discussion

In this study, we demonstrate that developmental activation of nociceptive C4da neurons in *Drosophila* larvae leads to sensitization of nociceptive behaviour. We show that this experience-dependent sensitization is accompanied by enhanced activity of C4da neurons in response to acute stimulation but occurs without significant changes in dendritic morphology. We identify the octopamine receptor OAMB in C4da neurons as a key molecular mediator of this sensitization, showing that its expression in C4da neurons is required for the enhancement of behavioural and neural responses. Furthermore, our data strongly suggest that OAMB in C4da neurons plays a role in experience dependent sensitization. We further identify VUM octopaminergic neurons as both anatomically and functionally connected to C4da neurons, and show that activation of VUMs is sufficient to induce sensitization. Finally, we provide evidence for a positive feedback loop between C4da neurons and VUMs, as prior nociceptive experience strengthens the functional connectivity between these populations, leading to increased responsiveness of both partners and amplification of nociceptive sensitization.

Our findings highlight a form of neuromodulator-driven plasticity within the *Drosophila* larvae nociceptive circuit in which prior noxious experience alters both the sensory neurons and their associated modulatory inputs. Specifically, stimulation of C4da nociceptive neurons during development leads to increased responsiveness in both C4da

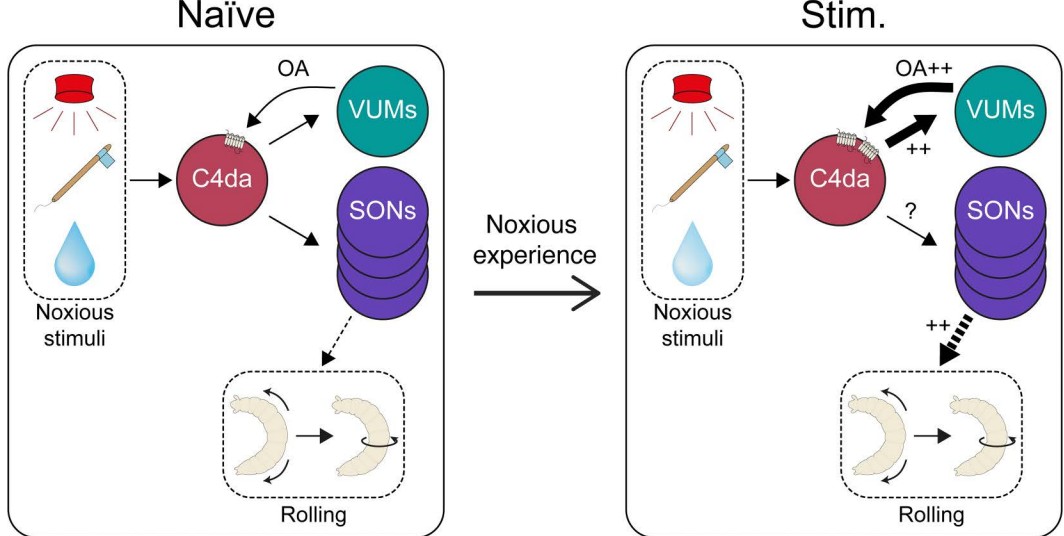

**Fig 7. Previous noxious experience leads to functional changes between C4da neurons and the octopaminergic VUMs.** In naïve larva (left), stimulation of C4da neurons activates the octopaminergic VUM through a direct synapse. VUM then sends feedback through octopamine, which is detected by OAMB in C4da, while other neurons trigger nocifensive rolling. Following previous noxious experience (right), C4da are sensitized and respond more strongly to a stimulus of identical strength. This, in turn, leads to increased activity in VUMs, which sends increased amounts of octopamine to C4da neurons. Together, these change increase escape response to noxious stimuli.

neurons through octopamine signaling and their upstream VUM modulatory neurons. These results suggest that the circuit adaptation involves changes at multiple nodes – enabling a more robust behavioural response to future noxious stimuli. This plasticity is experience-dependent and tightly linked to the internal state of the animal, supporting the idea that neuromodulatory systems can encode and store information about past sensory experiences to shape future behaviour [7,10–13,20,22,23]. Such mechanisms are reminiscent of central sensitization phenomena in vertebrates, where prior injury or repeated noxious input leads to persistent hypersensitivity and exaggerated behavioural responses [50,72]. Rather than being confined to a single locus, this form of plasticity appears to rely on reciprocal adjustments across sensory and modulatory components of the circuit. These bidirectional changes ensure that information about past noxious experiences is integrated into both receptor expression and circuit connectivity, supporting flexible yet precise tuning of nociceptive sensitivity that balances protective responsiveness with the need to avoid maladaptive overreaction.

Our MiMIC-gal4 experiment suggest that OAMB expression in C4da neurons is likely subject to dynamic regulation depending on the neuronal activation during development. Consistently, publicly available single-cell-RNA-seq datasets corroborates that OAMB is expressed at low levels in C4da neurons [73,74]. Further, upon closer examinations, we observe that sequences align either with the exon unique to -AS isoform, or with exons that are shared between both OAMB isoforms [73,74]. This observation aligns with our finding that overexpression of OAMB-AS in C4da neurons is sufficient to promote sensitization. Further study is required to clarify the experience-dependent expression of specific isoforms and the downstream intracellular pathways involved in this sensitization.

A growing body of evidence points toward a neuromodulatory and immune mechanisms by which *Drosophila* larvae dynamically regulate nociceptive behaviour based on the temporal pattern that is used to activate different classes nociceptive neurons [35,43,44]. Previous work has shown that serotonin contributes to behavioural habituation following persistent, high-frequency stimulation of C4da nociceptive neurons, suggesting a dampening mechanism engaged by sustained noxious exposure [43]. In contrast, recent results demonstrate that octopamine mediates sensitization, enhancing nociceptive responsiveness following intermittent, low-frequency developmental stimulation. Preliminarily, our data suggest that octopaminergic neurons are preferentially responsive to low-frequency input, while serotonergic neurons may be tuned to continuous, high-frequency stimulation. This dichotomy implies a frequency-dependent switch in neuromodulatory state that governs opposing behavioural outcomes—sensitization or habituation—depending on the nature of the threat. Such a mechanism could enable the larval nervous system to implement context-specific adaptations: brief, sporadic threats may trigger heightened vigilance via octopaminergic sensitization, while prolonged or inescapable stimuli may engage serotonergic habituation pathways to prevent maladaptive overreaction. This model reflects analogous processes in vertebrates, where distinct neuromodulatory systems mediate either pain amplification or suppression depending on stress duration and intensity [75]. By integrating frequency-dependent sensory coding with neuromodulatory feedback, *Drosophila* may employ a flexible system that adjusts behavioural responses to both immediate threat and cumulative sensory history—supporting a form of state-dependent plasticity in nociceptive processing (see *Upreti et al., 2019* [76], for a similar dual-process mechanism in *Aplysia*, involving structural plasticity in neuromodulatory circuits alongside synaptic changes between sensory and downstream neurons).

The identification of an octopaminergic feedback loop modulating nociceptive sensitivity in *Drosophila* larvae raises compelling questions about the evolutionary conservation of neuromodulatory control over pain and arousal states. Octopamine, the invertebrate analog of norepinephrine, serves many overlapping functions with its vertebrate counterpart—modulating arousal, locomotion, aggression, learning, and now, as we show, nociceptive plasticity. Our finding that the *Drosophila* OAMB receptor, a putative homolog of the vertebrate α1-adrenergic receptor [77], is both expressed in nociceptive sensory neurons and necessary for sensitization suggests that direct neuromodulatory input onto primary nociceptors may be a deeply conserved feature of bilaterian nervous systems. In mammals, peripheral adrenergic signaling has been implicated in pain hypersensitivity, with α1-adrenoceptors shown to enhance nociceptor excitability and contribute to

chronic pain states [78–81]. Similarly, the feedback loop we propose in *Drosophila*—where prior nociceptive experience amplifies neuromodulator release and receptor expression in nociceptors—parallels mechanisms observed in vertebrates, where injury or stress can lead to increased adrenergic tone and receptor upregulation in the periphery [78–83]. These similarities hint at a shared evolutionary strategy in which stress-related neuromodulators function as gain controls for sensory systems, tuning behavioural output to match the organism's internal state and environmental context. Our findings in *Drosophila* not only illuminate fundamental principles of nociceptive plasticity in invertebrates, but also provide a genetically tractable model to study how neuromodulatory systems interface with sensory circuits to shape adaptive—and potentially maladaptive—pain-related behaviours across species.

## Methods

### Fly stocks and maintenance

The fly stocks carrying the following genetic reagents were used in this study. The following stocks were obtained from the Bloomington Drosophila Stock Center (BDSC): UAS-GRAB(OA1.0) in attP40 (BDSC #604600); LexAop2-CsChrimson. tdTomato in VK00005 (BDSC #82183); UAS-IVS-CsChrimson-mVenus in attP18 (BDSC #55134); UAS-IVS-CsChrimson-mVenus in attP2 (BDSC #55136); LexAop-GAL80 in attp40 (BDSC #32214); GMR27H06-LexA in JK22C (BDSC #94664); GMR46B08-GAL4 in attP2 (BDSC #47361); LexAop-nSyb-spGFP1–10, UAS-CD4-spGFP11 (BDSC #64315); ppk-CD4:: tdTomato (BDSC #35844); tdc2-GAL4.S in attP2 (BDSC #52243); tdc2-lexA::p65 in attP40 (BDSC #52242); TRiP. HMS05829 in attP40 (BDSC #67968); TRIP.JF01673 in attP2 (BDSC #31171); TRIP.JF01732 in attP2 (BDSC #31233); TRIP.JF02746 in attP2 (BDSC #27667); UAS-Dcr2.D in 2 (BDSC #24650); UAS-GtACR1.d.EYFP in attP2 (BDSC #92983). The following stocks were kindly gifted by G. Rubin: UAS-IVS-mCD8::RFP in attP18; UAS-IVS-myr::GFP in attp2; UAS-Syn21-Chrimson88-tdTomato-3.1 in attP18; LexAop-CsChrimson-tdTomato in attP18; LexAop-mCD8::GFP in su(Hw)attP8; UAS-dsFRT-Chrimson in attp18; UAS-Syn21-opGCaMP6s in su(Hw)attP8; LexAop-FLP in attp40. Tsh-LexA and ppk1.9-LexA were kind gifts from J. Simpson. ppk1.9-GAL4 was kindly gifted by D. Tracey. UAS-IVS-GCaMP6s in attp2 was a kind gift from J. Jayaraman. Mi{Trojan-GAL4.1-Oamb}[MI12417-TG4.1] was a kind gift from R. S. Stowers. UAS-OAMB-AS and UAS-OAMB-K3 were kind gifts from D. J. Anderson. Information about the genetic constructs carried for experimental purposes can be found in S2 Table.

All *D. melanogaster* lines used in this study were maintained on standard cornmeal medium (Bloomington Drosophila Stock Center formulation) in a temperature- and humidity-controlled incubator at 25 ºC under a 12h light-dark cycle. For optogenetic experiments, flies and larvae were reared on same medium supplemented with 0.2 mM of all-trans retinal (Toronto Research Chemicals, R240000), prepared fresh and protected from light.

### Developmental optogenetic stimulations

Embryos were collected every 24 hours on food plates supplemented with all-trans retinal and larvae were reared in the dark until the third instar stage in the dark. To activate neurons during development, larvae expressing CsChrimson were exposed to pulsed red light stimulation (wavelength - 620 nm). Larvae were housed under LED array programmed to deliver five seconds pulses every five minutes at a power density of 3.53 µW/mm$^2$ unless otherwise noted. Light delivery was controlled via Raspberry Pi running on Raspberry OS, using a Python script controlled the array with the pigpio library general-purpose input-output control.

### Octopamine treatment

To manipulate octopaminergic signaling, larvae were fed octopamine hydrochloride (≥95% purity, Sigma Aldrich), following protocols adapted from *Saraswati et al., 2004* [18] and *Chen et al., 2013* [65]. Briefly, molten cornmeal agar was supplemented with red food coloring and octopamine at indicated concentrations. Two hours prior to the behavioural testing,

larvae were transferred to octopamine-containing plates or dye-only control plates. Only larvae that exhibited visible red coloration in the abdomen–indicating ingestion of the octopamine-enriched food–were selected for testing.

## Behavioural assays

Prior to behavioural testing, larvae were separated from the food medium using a 15% sucrose solution, transferred to a mesh sieve, and gently rinsed with distilled water. Larvae were then transferred to the appropriate behavioural arena, as described below. All behavioural manual annotation were conducted blind to genotype to minimize bias.

### Mechanical nociception

Mechanical nociception assays were conducted as described in *Zhu, Boivin et al., 2023* [84], as adapted from *Hu et al., 2017* [33]. Briefly, groups of 5–7 third-instar larvae were placed on humidified petri dish. Two subsequent mechanical stimuli were delivered to the dorsal side of each larva (segments A4-A6) using calibrated custom-made Von Frey filament (20-150mN). Behavioural responses were scored according to predefined ethogram on scale from 1 to 5, with 1 indicating no observable response and 5 indicating full rolling.

### Chemical nociception

Chemical nociception was assessed using methods adapted from *Lopez-Bellido et al., 2019* [25]. Groups of 5–7 larvae were placed on a petri dish and briefly dried with a Kimwipe. Individual larvae were then exposed to 1.5 μl of hydrochloric acid (HCl; 0.5%-10% concentration) applied to the tail using a pipette. Response latency was recorded, with the first complete rolling behaviour considered the response. Larvae failing to roll within 10 s were classified as non-responders.

### Thermal nociception

Thermal nociception assays followed the protocol by *Zhong et al., 2010* [85]. Group of 5–7 larvae were placed on a humidified petri dish. A calibrated heat prob was applied to the dorsal side between segments A4 and A6. Response latency was measured as the time to the first complete rolling behaviour. Larvae that did not roll within 10 s were considered non-responders.

### Optogenetic apparatus and assays

Optogenetic assays were adapted from *Ohyama et al., 2013* [86], with modifications to the hardware configuration as described by Zhu et al., 2024 [40]. The experimental setup consisted of a C-MOS Camera (BlackflyUSB3, BFLY-U3-23S6M-C, FLIR) with fixed focal length lens (Edmund, 56–529), 750 nm long-pass filter (Edmund, 66–575) and 850-nm infrared LED illumination for tracking, a 620-nm LED array for optogenetic stimulation, and a dedicated computer for stimulus delivery and data acquisition. Larval behaviour was recording and analysed using the Multi Worm Tracker (MWT), with stimulus delivery controlled via an integrated stimulus module.

For each trial, larvae were placed evenly on a 25 cm x 25 cm square agar plate (2% agar, pre-humidified using 15 ml of de-ionized water). The stimulation protocol consisted of two cycles of 30 s red-light stimulation (100 Hz, 90% duty cycle) interleaved with 30 s rest periods. Unless otherwise noted, experiments were conducted at a power density of 0.84 μW/mm$^2$.

### Behavioural data analysis

**Behaviour detection.** Behavioural feature extraction from optogenetic experiments was performed using the tracking data collected by the Multi-Worm Tracker (MWT) software. This data included larval contour, spine, and center of mass coordinates over time for individual animals. Kinematic parameters—including larval length, width, body area, speed,

crabspeed (lateral movement against body axis), body curvature, head angle, and directional bias—were computed using the Choreography software suite, as described in *Ohyama et al., 2013* [86], and in *Ohyama et al., 2015*, [30]. Larvae were included in the analysis only if they were continuously tracked for a minimum of 5 seconds and moved at least one body length. Collisions between larvae led to termination of tracking for the involved individuals, after which new object IDs were assigned.

Behavioural classification was made using machine-learning approaches applied to the features computed by the Choreography software. Non-nociceptive behaviours (i.e., crawling, hunching, stopping, and non-descript head casting) relied on a classifier developed and described by *Masson et al., 2020* [54]. Nociceptive behaviour, specifically rolling behaviour was detected using a classifier developed through the Janelia Automatic Animal Behaviour Annotator (JAABA) [30,53], trained using the rolling behaviour described in *Zhu et al., 2024* [40]. To improve the specificity of rolling detection, events lasting ≥0.5 s are considered as valid rolling behaviour.

**Behaviour quantification.** Detected behavioural events were aggregated and quantified using MATLAB scripts. Only larvae that were successfully tracked throughout the entire stimulation period were included in the analysis. For rolling behaviour, the following metrics were computed:

Rolling probability: the proportion of larvae that exhibited at least one rolling event during the stimulation period, relative to the total number of animals recorded.

Rolling latency: the time elapsed from the onset of the stimulus to the initiation of the first rolling event.

Total rolling time: the cumulative duration of all rolling events within the stimulation window.

Larvae that did not exhibit rolling behaviour during the stimulation were excluded from rolling latency and total duration analysis to avoid skewing comparisons due to differing response rate across groups.

## Gene expression analysis

For the validation of RNAi efficiency in larval neural tissues, RT-qPCR was conducted as described in *Das et al., 2025* [87], with *rp49* and *GAPDH* used as reference genes for the computation of the relative fold change in gene expression. Primers for OAMB and tbh were selected from the Fly Primer Bank. Selected primers target all isoform of the gene of interest and avoid regions that overlapped with RNAi to minimize the likelihood of a false positive.

OAMB: 5'-3' forward ATGAATGAAACAGAGTGCGAGG 5'-3' reverse GGCCAGGGAGATCAGATTGG; tbh: 5'-3' forward CTGAGCAGTCAGGATGGCATT 5'-3' reverse TGTGATGATGATAAGCCAGTTGG

**Larval dissections and immunohistochemistry.** Dissections and immunohistochemistry were conducted using standard protocols as described by *Patel, 1994* [88]. Briefly, either larval CNS or larvae fillet with intact peripheral nerve system (PNS) and CNS were dissected in phosphate-buffered saline (PBS). Tissues were then fixed in 4% paraformaldehyde (PFA) in PBS for 20 minutes at room temperature (RT), followed by three washes in PBS three times and two washes in PBS containing 0.4% Triton-X (PBX). Samples were then blocked in PBX supplemented with 5% normal goat serum (NGS) for 1 hour at RT. Following blocking, samples were incubated in primary antibodies for 1 hour at RT and subsequently overnight at 4°C. The following primary antibodies were used in PBX: chicken anti-GFP (1:3000, ab13970, Abcam), mouse anti-GFP (1:500, G6539, Sigma), rabbit anti-dsRed (1:1000, 632496 Takara/Clontech), rabbit anti-tdc2 (1:2000, ab128225, Abcam), mouse anti-nc82 (1:50, DHSB), rat anti-elav (1:50, DHSB). After primary antibody incubation, samples were washed six times in PBX (15 minutes each) at RT. Samples were then incubated in appropriate secondary antibodies (1:500, goat-anti-chicken/AF488(A11039), goat-anti-mouse/AF488(A21221), goat-anti-mouse/AF568(A11031), goat-anti-rabbit/AF568(A11011), goat-anti-rat/AF647(A21236), Thermo Fisher) an overnight at 4°C. Following secondary antibody incubation, samples underwent 6 additional 15-miniute PBS washes were mounted in VECTASHIELD anti-fade mounting medium. Samples were imaged using a Zeiss 710 LSM confocal microscope with a 20×/NA0.8 objective. Both qualitative and quantitative imaging were performed with identical laser power and gain setting within a single acquisition session to ensure consistency across samples. All image processing and quantification were conducted using FIJI (ImageJ).

## Immunohistochemistry image analyses

Quantitation of OAMB expression in C4da neurons was conducted in FIJI using Trojan-GAL4-driven GFP expression. For each neuron, z-stack comprising 2–3 optical sections were projected into a single maximum intensity image to capture the full volume of the sensory neurons cluster. A region of interest (ROI) was defined via thresholding in CsChrimson-tdTomato channel (red), and any non-specific signal was excluded manually before mask application. Mean fluorescence intensity values for GFP and Elav were measured within the ROI. To normalize across samples, GFP signal intensities were divided by the corresponding Elav mean intensity within the same ROI.

## Two-photon live imaging

**Sample preparation.** Live imaging of C4da dendritic activity was performed in intact larvae. Larvae were cold-anesthetized and mounted directly onto a glass slide. After drying the surface briefly, larvae were gently immobilized between the slide and a 50x24 mm coverslip to restrict movement, ensuring that dorsal cuticle remained stable and oriented toward the objective lens. Any larvae exhibiting movement during imaging were excluded from further analysis.

For imaging octopaminergic neuron activity, isolated CNS from third instar larvae were prepared following the protocol described by *Zhu et al., 2024* [40]. Briefly, the CNS was dissected in cold Baines physiological solution, then transferred onto a poly-L-lysine-coated cover glass shard for stabilization.

**Image acquisition and analysis.** Functional calcium imaging was conducted using a custom-build two-photon microscope equipped with Galvo-Resonant Scanner (Cambridge technology), operated by ScanImage software (MBF Bioscience). A $40 \times /0.80$NA water immersion objective (LUMPlanFL, Olympus) was used for image acquisition. GCaMP was excited at 920 nm using a Mai Tai ultrafast pulsed laser (Spectra-Physics). Emission signals were detected by GaAsP photomultiplier tubes (Hamamatsu) controlled by a HHMI PMT controller. Images were acquired at 30 frames per second (fps) on a single focal plane with spatial resolution of 512 x 512 pixels. A 620-nm LED (Thorlabs) was used for optogenetic stimulation. Stimuli were delivered in three repeated trials, with the following conditions: Whole larvae GCaMP6s imaging: 10 pulses (30 ms each) over 1,000 ms, repeated every 30 s. CNS GCaMP6s imaging: 100 pulses (9 ms each) over 1,000 ms, repeated every 60 s. GRAB-OA1.0 imaging: 500 pulses (9 ms each) over 5,000 ms, repeated every 60 s. Unless otherwise noted, the stimulation intensity was set to 0.1 $\mu$W/mm$^2$.

To reduce background noise, acquired images were averaged to yield a time-lapse stack with a 10-fps temporal resolution. Image analysis was performed using FIJI and MATLAB. ROI was defined based on standard deviation projections across full time recording. Fluorescence intensity was quantified as $\Delta F/F_0$, where $\Delta F = F - F_0$ and $F_0$ represents the mean baseline fluorescence during the 5 seconds preceding stimulus onset. Background fluorescence was subtracted prior to normalization. Responses were then averaged across the three stimulations trials per preparation. The following metrics were extracted from resulting traces: peak before post-processing to obtain the peak $\Delta F/F_0$, latency to peak, and signal decay time as the represented as the half-life from the peak.

## Statistics

The JMP 18 Pro software (SAS Institute Inc.) was used for statistical analysis. For continuous data types, the Shapiro-Wilk test was used to validate the normality of the dataset. If the assumption is respected, the groups were compared using a one-way ANOVA, with post-hoc Fisher's LSD test used to determine pairwise comparison in cases of multiple comparisons. Welch's ANOVA was used in case of severe discrepancies in n or unequal variance as asserted with Levene's test. Continuous data that failed to meet assumptions and ordinal data led to the use of a non-parametric equivalent, namely the Kruskal-Wallis test followed by a non-parametric Fisher's LSD test based on pairwise Mann-Whitney test, or a Mann-Whitney test for comparisons across two samples. Categorical data were analyzed using pairwise Chi-square tests, with a Bonferroni correction in cases of multiple comparisons. The type of statistical test used in each experiment

is indicated in figure legends. Sample numbers are indicated in figure legends. P values are represented by asterisks: **: p<0.01, ***: p<0.001. For multiple comparisons, the compact letter display (CLD) is used at p<0.01. Summary values represented in figures can be found in S3 Table. Exact p-values and tests can be found in S4 Table.

## Supporting information

**S1 Fig. Key features of developmental stimulations differentially affect rolling behaviour.** (A-C) Acute nociceptive stimulations modestly affects response to subsequent stimulations. (A) Rolling probability in larvae depending on the stimulation number within the experiment. Bars show mean with 95% confidence intervals (n=235, 255). Chi-square test, **p<0.01 (B) Violin plot of rolling latency during optogenetic stimulation (n=117, 170). Kruskall-Wallis test, ***p<0.001. (C) Violin plot of total time each larvae spent rolling during optogenetic stimulations (n=117, 170). Kruskall-Wallis test. (D-F) Developmental stage has an incremental effect on sensitization, with most potent effects with developmental stimulation of late-third instar. (D) Rolling probability in larvae based on developmental time window of stimulation. Bars show mean with 95% confidence intervals (n=237, 234, 267, 187, 211, 283, 260). Chi-Square Test with compact letter display; groups not sharing a letter differ significantly at p<0.01 (E) Violin plot of rolling latency during optogenetic stimulation (n=106, 125 138, 113, 165, 229, 254). Kruskall-Wallis followed by pairwise Mann-Whitney test, CLD denotes p<0.01. (F) Violin plot of total time each larvae spent rolling during optogenetic stimulations (n=106, 125 138, 113, 165, 229, 254). Kruskall-Wallis followed by pairwise Mann-Whitney test; CLD denotes p<0.01. (G-I) Stimulation intensity increases sensitization, but the effect rapidly saturates. (G) Rolling probability during optogenetic C4da stimulation. Error bars: 95% confidence interval (n=517, 208, 185, 160, 124, 214). Chi-square test with Bonferroni correction; CLD denotes p<0.01. (H) Violin plot of rolling latency during optogenetic stimulation (n=258, 125, 180, 160, 121, 208). Kruskall-Wallis followed by pairwise Mann-Whitney test, CLD denotes p<0.01. (I) Violin plot of total time each larvae spent rolling during optogenetic stimulations (n=258, 125, 180, 160, 121, 208). Kruskall-Wallis followed by pairwise Mann-Whitney test; CLD denotes p<0.01.
(TIF)

**S2 Fig. Developmental stimulations do not affect C4da dendrite morphology.** (A-A') Representative dendritic arbour from naïve and developmentally stimulated larvae. Scale bar=50 μm. (A) Naïve. (A') Stimulated. (B) Average Sholl profiles for naïve (n=7) and stimulated larvae (n=8). Profiles were normalized to control for inter-individual variance in larval size. (C-E) Boxplots representing different core features of the normalized Sholl profiles in naïve (n=7) and stimulated larvae (n=8). (C) Area under the curve. (D) Critical radius. (E) Number of intersection at critical radius.
(TIF)

**S3 Fig. The octopamine receptor OAMB is required for experience-dependent sensitization to noxious stimuli.** (A-C) An RNAi screening identifies OAMB in C4da neurons as critical for sensitization following noxious experience. (A) Rolling probability during optogenetic C4da stimulation. Error bars: 95% confidence interval (n=100, 100, 100, 100, 100, 93, 100, 100, 100, 100, 100, 100, 100, 100, 100, 70, 100, 100, 100). Chi-square test; statistical marks are relative to the No RNAi/ Stim. group. (B) Violin plot of rolling latency during optogenetic stimulation (n=48, 85, 70, 80, 62, 58, 74, 74, 86, 37, 13, 69, 96, 65, 51, 70, 98, 70, 86). Kruskall-Wallis followed by pairwise a Steel-Dwass test with control (No RNAi/ Stim.) (C) Violin plot of total time each larvae spent rolling during optogenetic stimulations (n=48, 85, 70, 80, 62, 58, 74, 74, 86, 37, 13, 69, 96, 65, 51, 70, 98, 70, 86). Kruskall-Wallis followed by pairwise a Steel-Dwass test with control (No RNAi/ Stim.). (D) Relative fold-change in OAMB expression following whole-brain RNAi knockdown. Error bars represent SEM for biological replicates (n=3). Kruskall-Wallis followed by pairwise Mann-Whitney test; CLD denotes p<0.01. (E-F) Calcium imaging of C4da neurons in axon terminal during optogenetic stimulation with or without OAMB RNAi. n=12, 13, 12, 13. (E) $\Delta F/F_0$ traces from individual neurons aligned to stimulus onset. Shaded region indicates stimulation period. (F) Maximum $\Delta F/F_0$ responses in naïve and stimulated larvae with or without RNAi. Mann-Whitney test; CLD indicates

statistical significance at $p < 0.01$. (G-H) OAMB expression varies with neuronal identity. (G) Representative images mean OAMB expression in vdaB neurons in naïve and stimulated larvae. Scale bar = 10 µm. (H) Representative images mean OAMB expression in v'ada neurons in naïve and stimulated larvae. Scale bar = 10 µm. (I-K) Overexpression of the OAMB-AS isoform, but not the OAMB-K3 isoform, phenocopies the effect of developmental stimulations. (I) Rolling probability in larvae depending on expression of different OAMB isoforms. Bars show mean with 95% confidence intervals (n = 207, 289, 238). Chi-Square Test with compact letter display; groups not sharing a letter differ significantly at $p < 0.01$ (J) Violin plot of rolling latency during optogenetic stimulation (n = 110, 139, 214). Kruskall-Wallis followed by pairwise Mann-Whitney test, CLD denotes $p < 0.01$. (K) Violin plot of total time each larvae spent rolling during optogenetic stimulations (n = 110, 139, 214). Kruskall-Wallis followed by pairwise Mann-Whitney test; CLD denotes $p < 0.01$.
(TIF)

**S4 Fig. Octopaminergic signalling from tdc2 + neurons affects larval rolling behaviour following developmental stimulations.** (A-B) RNAi knockdown of the rate-limiting enzyme tbh affects rolling latency and total duration. (A) Violin plot of rolling latency during optogenetic stimulation (n = 44, 83, 55, 70, 292, 226, 163, 167). Kruskall-Wallis followed by pairwise Mann-Whitney test, CLD denotes $p < 0.01$. (B) Violin plot of total time each larvae spent rolling during optogenetic stimulations (n = 44, 83, 55, 70, 292, 226, 163, 167). Kruskall-Wallis followed by pairwise Mann-Whitney test; CLD denotes $p < 0.01$. (C) Relative fold-change in tbh expression following whole-brain RNAi knockdown. Error bars represent SEM for biological replicates (n = 3). Kruskall-Wallis followed by pairwise Mann-Whitney test; CLD denotes $p < 0.01$. (D-E) Optogenetic silencing during developmental stimulations affects rolling latency but not total duration (D) Violin plot of rolling latency during optogenetic stimulation (n = 51, 263, 144, 76, 265, 217). Kruskall-Wallis followed by pairwise Mann-Whitney test, CLD denotes $p < 0.01$. (E) Violin plot of total time each larvae spent rolling during optogenetic stimulations (n = 51, 263, 144, 76, 265, 217). Kruskall-Wallis followed by pairwise Mann-Whitney test; CLD denotes $p < 0.01$.
(TIF)

**S5 Fig. Combinations of genetic reagents allow for isolation different tdc2 + neuron clusters.** (A-D) Example images of different techniques used to drive CsChrimson in different tdc2 + neuron clusters. Scale bar = 50 µm. (A'-D') Individual tdc2 + thoracic/abdominal neuronal population (i.e., DUM, VPM, VUM) in each isolation techniques. Scale bar = 20 µm. (A-A') Expression of UAS-CsChrimson driven by tdc2-GAL4. (B-B') Expression of UAS-CsChrimson driven by tdc2-GAL4 combined to GAL80 expression driven by tsh-LexA, allowing for isolation of octopaminergic neurons of the brain. (C-C') Expression of UAS-FRT.Stop-CsChrimson driven by tdc2-GAL4 combined to FLP expression driven by tsh-LexA, allowing for isolation of octopaminergic neurons of the ventral nerve chord. (D-D') Expression of UAS-CsChrimson driven by 46B08-GAL4, allowing for isolation of the VUMs.
(TIF)

**S6 Fig. The locus for frequency-encoding towards habituation does not lie within tdc2 neurons.** (A-C) Calcium imaging of tdc2 + VUM neurons reveals a small change in temporal dynamics under an habituation paradigm (A) GCaMP6s traces of individual and average response in VUM neurons; shaded region indicates stimulus. (B) Peak $\Delta F/F_0$ in naïve vs. stimulated larvae (n = 12, 10). (C) Time to reach the peak $\Delta F/F_0$ in naïve vs. stimulated larvae (n = 12, 10). Mann-Whitney test ** $p < 0.01$. (D) Stimulation of tdc2 + neurons does not phenocopies stimulation of C4da neurons under an habituation paradigm. Violin plot of latency to roll following application of 5% HCl (n = 52, 53, 62, 50, 51, 46).
(TIF)

**S1 Movie. Representative recording of GCaMP6s signal in the C4da dendritic field upon its optogenetic activation for a naïve larva.**
(AVI)

**S2 Movie. Representative recording of GCaMP6s signal in the C4da dendritic field upon its optogenetic activation for a stimulated larva.**
(AVI)

**S1 Table. Number of hemisegments in which C4da neurons frorm synapses with tdc2+neurons.**
(XLSX)

**S2 Table. The flies carrying the following constructs were used for experiments and were represented in figures as follows.**
(XLSX)

**S3 Table. Summary values for the data represented in figures.**
(XLSX)

**S4 Table. Summary of statistical analysis per figure.**
(XLSX)

## Acknowledgments

Confocal images were collected at the McGill University Advanced Bio Imaging Facility (ABIF), RRID:SCR_017697. We thank for Bloomington stock center for providing the fly stocks. We thank V. Jayaraman, G. Rubin, J. Simpson, R. S. Stowers, D. Tracey and D. J. Anderson for sharing their fly stocks. We thank Arghya Das and Nam-Sung Moon for their assistance with the RT-qPCR. We thank Marta Zlatic and members of Ohyama lab for critical comments on the manuscript.

## Author contributions

**Conceptualization:** Jean-Christophe Boivin, Tomoko Ohyama.

**Data curation:** Jean-Christophe Boivin, Yi Q. Zhao, Jiayi Zhu, Jared T. Dakin, Jing Ning, Tomoko Ohyama.

**Formal analysis:** Jean-Christophe Boivin, Tomoko Ohyama.

**Funding acquisition:** Tomoko Ohyama.

**Supervision:** Tomoko Ohyama.

**Writing – original draft:** Jean-Christophe Boivin, Tomoko Ohyama.

**Writing – review & editing:** Jean-Christophe Boivin, Yi Q. Zhao, Tomoko Ohyama.

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
