## [Decision Letter · Decision Letter 0]

3 Nov 2025

PGENETICS-D-25-00927

A positive feedback loop between sensory and octopaminergic neurons underlies nociceptive plasticity in Drosophila larvae

PLOS Genetics

Dear Dr. Ohyama,

Thank you for submitting your manuscript to PLOS Genetics. After careful consideration, we feel that it has merit but does not fully meet PLOS Genetics's publication criteria as it currently stands. Therefore, we invite you to submit a revised version of the manuscript that addresses the points raised during the review process.

Please submit your revised manuscript within 60 days Jan 02 2026 11:59PM. If you will need more time than this to complete your revisions, please reply to this message or contact the journal office at plosgenetics@plos.org. Please include the following items when submitting your revised manuscript:

We look forward to receiving your revised manuscript.

Kind regards,

Gaiti Hasan

Academic Editor

PLOS Genetics

Fengwei Yu

Section Editor

PLOS Genetics

Aimée Dudley

Editor-in-Chief

PLOS Genetics

Anne Goriely

Editor-in-Chief

PLOS Genetics

**Journal Requirements:**

At this stage, the following Authors/Authors require contributions: Jean-CHristope Boivin, Yi Q. Zhao, Jiayi Zhu, Jared T. Dakin, and Jing Ning. Please ensure that the full contributions of each author are acknowledged in the "Add/Edit/Remove Authors" section of our submission form.

The list of CRediT author contributions may be found here: https://journals.plos.org/plosgenetics/s/authorship#loc-author-contributions

https://journals.plos.org/plosgenetics/s/submission-guidelines#loc-parts-of-a-submission

4) In the online submission form, you indicated that "The original contributions presented in this study are included in the article/supplemental material, further inquiries can be directed to the corresponding author.". All PLOS journals now require all data underlying the findings described in their manuscript to be freely available to other researchers, either

1. In a public repository

2. Within the manuscript itself

3. Uploaded as supplementary information.

**Reviewers' comments:**

Reviewer's Responses to Questions

**Comments to the Authors:**

Reviewer #1: The authors report on experiments using Drosophila melanogaster larvae, in which they stimulated nociceptive sensory cells either mechanosensorically or optogenetically. They find that after prolonged stimulation, depending on frequency and intensity, the animals’ behavioral responses are either reduced or enhanced. They refer to these opposing processes as desensitization and sensitization, respectively, and attribute the sensitizing effect to the activity of specific octopaminergic (VUM) neurons. The study employs modern methodology (optogenetics, optical imaging) and several elegant technical approaches.

Overall, this is a well-conducted and informative study. The experimental design is mostly logical, sample sizes are appropriate, and statistical analyses are sound. The conclusions are clear and well supported by the data. I have only a few minor comments that the authors could address in a revision.

1. The terms desensitization and sensitization are typically defined very specifically. Desensitization usually refers to the reversal of a previously established sensitization. In the present case, the phenomenon described as “desensitization” would more appropriately be termed habituation. The authors should clarify this distinction and adjust the terminology accordingly throughout the manuscript.

2. The larvae display either habituation or sensitization depending on the duration, frequency, and intensity of stimulation. However, the main figures and text focus primarily on sensitization, with habituation data relegated to the supplementary materials. Since the switch between these behavioral states is particularly intriguing, I recommend presenting both phenomena in the main figures.

It would also be informative to clarify how VUM neuron activity relates to habituation (or desensitization) under high-frequency stimulation. Do the octopaminergic neurons act as a switch mechanism between these behavioral outcomes?

3. Figure 1B: The Y-axis label “larvae index” was confusing. Why is this expressed as an index rather than as the number of larvae? Clarifying this would help readers interpret the data correctly.

4. Figure S3 shows an RNAi screen targeting various GPCRs. The data for the OAMB receptor, which was analyzed in more detail, are convincing. However, the RNAi efficiency should be directly demonstrated, at least for OAMB. For the other RNAi lines, please note explicitly that in cases where no behavioral effect was observed, it remains unclear whether the receptor is nonessential or whether the RNAi line was ineffective. Similarly, for the RNAi-mediated knockdown of tdc, molecular evidence (e.g., qPCR or immunostaining) should be provided to confirm that the gene product is effectively downregulated.

5. There is a pressing need to improve the presentation and use of genetic controls for experiments involving RNAi lines. In some cases (e.g., Figure 3), only a “no RNAi” group is indicated as the contro (specified only in the supplementary table). However, it is standard practice—and essential for scientific rigor—to include both parental control genotypes in the analysis: the UAS-RNAi line crossed to a control background (without the driver), and the Gal4 driver line crossed to a control background (without the RNAi construct). Both heterozygous parental controls must be tested alongside the Gal4>UAS:RNAi experimental cross to rule out background or driver-related effects. Including these controls is not optional but required to meet current methodological standards in Drosophila genetics.

In summary, this is a strong and elegant study that provides valuable insight into the mechanisms of nociceptive plasticity in Drosophila larvae. With some clarifications and several additions, the manuscript will be even clearer and more comprehensive. I congratulate the authors on this successful and informative work.

Reviewer #2: The manuscript titled ‘A positive feedback loop between sensory and octopaminergic neurons underlies nociceptive plasticity in Drosophila larvae’ by Boivin et al. investigates how developmental activation of nociceptive C4da neurons in Drosophila larvae drives experience-dependent sensitization of nociceptive behavior. The work makes three major advances: (1) It establishes that sensitization arises without morphological remodeling of C4da dendrites but instead through enhanced neuronal activity, pointing to functional rather than structural mechanisms of adaptation. (2) It identifies the octopamine and its receptor OAMB as a critical modulator of sensitization. (3) It suggests direct connectivity between C4da neurons and the VUM octopaminergic motor neurons (such direct sensory neuron-to-motor neuron connections are thought to be rare), and demonstrates that VUM activation is sufficient to induce sensitization of behavior. Together, these findings highlight a neuromodulator-driven mechanism of plasticity that tunes nociceptive sensitivity and behavioral output according to developmental experience, paralleling vertebrate central sensitization and pointing to an evolutionarily conserved strategy in which adrenergic/octopaminergic signaling dynamically calibrates nociceptive circuits.

The manuscript, including experiments, are well considered and we request only minor revisions. These suggestions are listed below in the order of the text.

Line 110: “optogenetic” should be revised to “optogenetically”.

Line 111: “specially” should be revised to “especially”.

Lines 115-116: “, with the” should be revised to “occuring”.

Line 118: “spend” should be revised to “spent”

Figure 1B: please label the conditions.

Figure 3D: please expand the figure caption to explain all the symbols. Alternatively, simplify the schematic to show each sensory neuron using the same symbol and highlight just the C4da neurons in red.

Figure 3E: calcium imaging is shown only for ddaC but the caption states it is shown for all C4da neurons.

Mentioning earlier in the text that C4da neurons is a class of sensory neurons that comprises three neurons may improve readability when the manuscript switches from considering C4da neurons as a whole versus individual neurons.

Figure 3G: consider adding a non C4da neuron here as a negative control.

Figure 4A: why do the UAS-tbh-RNAi(1) and UAS-tbh-RNAi(2) control lines differ significantly in their rolling probability? Please explain.

Figure 4C: please plot UAS-GtACR1 controls as well.

Figure 6A: explain what “nc82” stains for in the figure caption. Add contextual details about the view and location of the slices that have been shown.

Add a supplementary figure showing the absence of co-staining between tdc2+ neurons and the other octopaniergic neurons (i.e., DUMs, ABLKs, and VPMs).

Consider confirming the synaptic connections between the VUMs and C4da neurons in the larval Drosophila melanogaster VNC connectome and including these data as a figure panel.

We recommend adding a summary schematic of the proposed circuit to Figure 6.

Figures S2 should be presented before S3 in the supplementary document.

Table S1 caption should say “from” rather than “form”.

Reviewer #3: The study by Boivin et al aims to define the cellular, molecular and circuit-level mechanisms of a form of adaptive modulation of nociceptive behavior in Drosophila. The authors use a combination of optogenetic and naturalistic behavior paradigms to convincingly demonstrate that chronic optogenetic stimulation drives lasting sensitization to nociceptive inputs. They further show that this sensitization involves heightened activation of nociceptive neurons to noxious inputs, requires octopamine receptor function in nociceptors, and identify neurons in the VNC that participate in the sensitization. Overall this is a well-designed study and results are generally clear and convincing (particularly behavioral evidence and calcium imaging demonstrating sensitization), though the physiological relevance of the phenomenology is not clear (point 1 below). The proposed mechanism of sensitization (increased OAMB expression) is not supported by the results provided, but this is not an essential element of the manuscript. Finally, the model for feed-forward control in the circuit is intriguing as this appears to represent an invertebrate model for central sensitization to noxious inputs, however this requires further substantiation (see points 3 and 4 below).

1. The authors present compelling evidence that chronic optogenetic nociceptor stimulation drives nociceptive sensitization that persists for hours or days. However, the paradigms all rely on optogenetic stimulation for extended periods (24 h or more) to trigger the mechanism under study. The study would be substantially strengthened by identification of roles for this octopaminergic feed-forward mechanism in some physiologically relevant situation.

2. The evidence that OAMB-R expression is responsive to nociceptor activation is not compelling. The studies with the transcriptional reporter suggest that OAMB-R levels might change in ddaC in response to activation, but no evidence is provided to evaluate whether modulating OAMB-R levels in ddaC is sufficient to induce this form of sensitization. Calcium imaging studies show that the enhanced calcium responses of ddaC to optogenetic stiumuls are attenuated by OAMB RNAi – do v’ada and At a minimum, the authors show use more direct assays to monitor OAMB-R RNA expression levels (in situs) to support their claims. Does acute OAMB expression (for example using the gene-switch system) sensitize nociceptors to noxious inputs?

3. The authors present syb-GRASP results that are suggestive of synaptic connectivity between C4da neurons and VUMs. Their calcium imaging results demonstratethat VUM responds to C4da activation and that prior C4da stimulation heightens VUM responses to VUM activity, however their studies as presented do not support the conclusions that prior activation of C4da neurons increases synaptic connectivity to VUM.

4. The authors discuss a role for both pre- and post-synaptic changes in mediating sensitization. What is the evidence for post-synaptic changes in VUM neurons following prior stimulation of C4da neurons? In figure 6, the authors presesnt imaging studies showing that octopamine release onto C4da axons increases following prior octopaminergic stimulation. Do they see a similar effect to prior C4da stimulation? They should include this experiment as it provides a more compelling test of their model.

Minor

1. The details of the paradigm in 1F are not completely clear. Larvae are being stimulated in the indicated developmental window, but when is behavior being assayed? If they’re all being assayed at the same time (120-144 h AEL?), the results don’t necessarily support the conclusion that stimulation of older larvae generates more substantial potentiation if the effects decay after removal of the conditioning stimulus.

2. The study demonstrates that Octopamine feeding acutely sensitizes larvae to noxious stimuli. This result seems at odds with the model that OAMB induction plays an important role in sensitization.

3. Related to the octopamine feeding experiments. The authors should include cell-specific OAMB knockdown in C4da neurons to rule out the possible contribution of octopamine signaling through other da neurons to the observed responses. Does OAMB RNAi in C4da neurons block effects of octopamine feeding?

4. VUM stimulation drives only a modest increase in nociceptive behavioral output (~38% response for controls, ~50-52% response for tdc2/vum>chrimson; prior C4da stimulation or OA treatment yielded ~80% response in these assays. Why the modest sensitization to vum stimulation? The plots would be easier to follow if the naive and pre-stimulated conditions are shown together for each genotype

5. The authors overstate the effects of octopaminergic silencing on the C4da sensitization. Their results (Fig. 4C) are consistent with tdc2-GAL4 neurons contributing to sensitization, but other neurons appear to contribute as well

**Have all data underlying the figures and results presented in the manuscript been provided?**

Large-scale datasets should be made available via a public repository as described in the *PLOS Genetics*
data availability policy, and numerical data that underlies graphs or summary statistics should be provided in spreadsheet form as supporting information., and numerical data that underlies graphs or summary statistics should be provided in spreadsheet form as supporting information.

Reviewer #1: **No:** I did not find any link to get access to the original data.I did not find any link to get access to the original data.

Reviewer #2: Yes

Reviewer #3: Yes

PLOS authors have the option to publish the peer review history of their article (what does this mean?). If published, this will include your full peer review and any attached files.). If published, this will include your full peer review and any attached files.

.

Reviewer #1: No

Reviewer #2: No

Reviewer #3: No

**Figure resubmission:**
---

## [Decision Letter · Decision Letter 1]

3 Mar 2026

PGENETICS-D-25-00927R1

A positive feedback loop between sensory and octopaminergic neurons underlies nociceptive plasticity in Drosophila larvae

PLOS Genetics

Dear Dr. Ohyama,

Thank you for submitting your manuscript to PLOS Genetics. After careful consideration, we feel that it has merit but does not fully meet PLOS Genetics's publication criteria as it currently stands. Therefore, we invite you to submit a revised version of the manuscript that addresses the points raised during the review process.

Please submit your revised manuscript within by Apr 02 2026 11:59PM. If you will need more time than this to complete your revisions, please reply to this message or contact the journal office at plosgenetics@plos.org. Please include the following items when submitting your revised manuscript:

We look forward to receiving your revised manuscript.

Kind regards,

Gaiti Hasan

Academic Editor

PLOS Genetics

Fengwei Yu

Section Editor

PLOS Genetics

Aimée Dudley

Editor-in-Chief

PLOS Genetics

Anne Goriely

Editor-in-Chief

PLOS Genetics

**Additional Editor Comments:**

One of the reviewers has a minor point they would like you to address.

**Journal Requirements:**

Please ensure that the CRediT author contributions listed for every co-author are completed accurately and in full.

At this stage, the following Authors/Authors require contributions: Jean-Christope Boivin, Yi Q. Zhao, Jiayi Zhu, Jared T. Dakin, and Jing Ning. Please ensure that the full contributions of each author are acknowledged in the "Add/Edit/Remove Authors" section of our submission form.

The list of CRediT author contributions may be found here: https://journals.plos.org/plosgenetics/s/authorship#loc-author-contributions

**Reviewers' comments:**

Reviewer's Responses to Questions

**Comments to the Authors:**

Reviewer #1: The authors have done a great job in addressing/clarifying my points of concern. I recommend acceptance of the manuscript.

Reviewer #3: The authors have been responsive to reviewers comments and substantially improved the manuscript. I still have some questions about the connection to OAMB. The authors show that OAMB RNAi attenuates calcium responses in ddaC (but not other class IV da neurons - why??) in response to optogenetic stimulation and provide additional studies showing that chronic overexpression of one isoform induces sensitization. However, the authors do not provide any direct evidence that the particular isoform is expressed in class IV da neurons, or that OAMB expression is responsive to activity (the GAL4 insertion is used as a proxy but it's not clear that this faithfully reports on the expression). Given the centrality of the receptor to the pathway, this seems like a point that should be addressed.

**Have all data underlying the figures and results presented in the manuscript been provided?**

Large-scale datasets should be made available via a public repository as described in the *PLOS Genetics*
data availability policy, and numerical data that underlies graphs or summary statistics should be provided in spreadsheet form as supporting information., and numerical data that underlies graphs or summary statistics should be provided in spreadsheet form as supporting information.

Reviewer #1: Yes

Reviewer #3: None

PLOS authors have the option to publish the peer review history of their article (what does this mean?). If published, this will include your full peer review and any attached files.). If published, this will include your full peer review and any attached files.

.

Reviewer #1: No

Reviewer #3: No

**Figure resubmission:**
---

## [Editor Report · Decision Letter 2]

6 Apr 2026

Dear Dr Ohyama,

We are pleased to inform you that your manuscript entitled "A positive feedback loop between sensory and octopaminergic neurons underlies nociceptive plasticity in Drosophila larvae" has been editorially accepted for publication in PLOS Genetics. Congratulations!

Yours sincerely,

Gaiti Hasan

Academic Editor

PLOS Genetics

Fengwei Yu

Section Editor

PLOS Genetics

Aimée Dudley

Editor-in-Chief

PLOS Genetics

Anne Goriely

Editor-in-Chief

PLOS Genetics

BlueSky: @plos.bsky.social

Comments from the reviewers (if applicable):

**Data Deposition**

If you have submitted a Research Article or Front Matter that has associated data that are not suitable for deposition in a subject-specific public repository (such as GenBank or ArrayExpress), one way to make that data available is to deposit it in the Dryad Digital Repository. As you may recall, we ask all authors to agree to make data available; this is one way to achieve that. A full list of recommended repositories can be found on our . As you may recall, we ask all authors to agree to make data available; this is one way to achieve that. A full list of recommended repositories can be found on our website..

http://datadryad.org/submit?journalID=pgenetics&manu=PGENETICS-D-25-00927R2

Additionally, please be aware that our data availability policy requires that all numerical data underlying display items are included with the submission, and you will need to provide this before we can formally accept your manuscript, if not already present. requires that all numerical data underlying display items are included with the submission, and you will need to provide this before we can formally accept your manuscript, if not already present.

**Press Queries**

If you or your institution will be preparing press materials for this manuscript, or if you need to know your paper's publication date for media purposes, please inform the journal staff as soon as possible so that your submission can be scheduled accordingly. Your manuscript will remain under a strict press embargo until the publication date and time. This means an early version of your manuscript will not be published ahead of your final version. PLOS Genetics may also choose to issue a press release for your article. If there's anything the journal should know or you'd like more information, please get in touch via plosgenetics@plos.org..

---

## [Editor Report · Acceptance letter]

PGENETICS-D-25-00927R2

A positive feedback loop between sensory and octopaminergic neurons underlies nociceptive plasticity in Drosophila larvae

Dear Dr Ohyama,

We are pleased to inform you that your manuscript entitled "A positive feedback loop between sensory and octopaminergic neurons underlies nociceptive plasticity in Drosophila larvae" has been formally accepted for publication in PLOS Genetics! Your manuscript is now with our production department and you will be notified of the publication date in due course.

With kind regards,

Anita Estes

PLOS Genetics

On behalf of:
